# ELUCIDATING THE EXPOSURE BIAS IN DIFFUSION MODELS

**Mang Ning**
Utrecht University
m.ning@uu.nl

**Mingxiao Li**
KU Leuven
mingxiao.li@cs.kuleuven.be

**Jianlin Su**
Moonshot AI Ltd.
bojone@spaces.ac.cn

**Albert Ali Salah**
Utrecht University
a.a.salah@uu.nl

**Itir Onal Ertugrul**
Utrecht University
i.onalertugrul@uu.nl

## ABSTRACT

Diffusion models have demonstrated impressive generative capabilities, but their *exposure bias* problem, described as the input mismatch between training and sampling, lacks in-depth exploration. In this paper, we investigate the exposure bias problem in diffusion models by first analytically modelling the sampling distribution, based on which we then attribute the prediction error at each sampling step as the root cause of the exposure bias issue. Furthermore, we discuss potential solutions to this issue and propose an intuitive metric for it. Along with the elucidation of exposure bias, we propose a simple, yet effective, training-free method called Epsilon Scaling to alleviate the exposure bias. We show that Epsilon Scaling explicitly moves the sampling trajectory closer to the vector field learned in the training phase by scaling down the network output, mitigating the input mismatch between training and sampling. Experiments on various diffusion frameworks (ADM, DDIM, EDM, LDM, DiT, PFGM++) verify the effectiveness of our method. Remarkably, our ADM-ES, as a state-of-the-art stochastic sampler, obtains 2.17 FID on CIFAR-10 under 100-step unconditional generation. The code is at https://github.com/forever208/ADM-ES

## 1 INTRODUCTION

Due to the outstanding generation quality and diversity, diffusion models (Sohl-Dickstein et al., 2015; Ho et al., 2020; Song & Ermon, 2019) have achieved unprecedented success in image generation (Dhariwal & Nichol, 2021; Nichol et al., 2022; Rombach et al., 2022; Saharia et al., 2022), audio synthesis (Kong et al., 2021; Chen et al., 2021) and video generation (Ho et al., 2022). Unlike generative adversarial networks (GANs) (Goodfellow et al., 2014), variational autoencoders (VAEs) (Kingma & Welling, 2014) and flow-based models (Dinh et al., 2014; 2017), diffusion models stably learn the data distribution through a noise/score prediction objective and progressively removes noise from random initial vectors in the iterative sampling stage.

A key feature of diffusion models is that good sample quality requires a long iterative sampling chain since the Gaussian assumption of reverse diffusion only holds for small step sizes (Xiao et al., 2022). However, Ning et al. (2023) claim that the iterative sampling chain also leads to the *exposure bias* problem (Ranzato et al., 2016; Schmidt, 2019). Concretely, given the noise prediction network $\epsilon_{\boldsymbol{\theta}}(\cdot)$, exposure bias refers to the input mismatch between training and inference, where the former is always exposed to the ground truth training sample $\boldsymbol{x}_t$ while the latter depends on the previously generated sample $\hat{\boldsymbol{x}}_t$. The difference between $\boldsymbol{x}_t$ and $\hat{\boldsymbol{x}}_t$ causes the discrepancy between $\epsilon_{\boldsymbol{\theta}}(\boldsymbol{x}_t)$ and $\epsilon_{\boldsymbol{\theta}}(\hat{\boldsymbol{x}}_t)$, which leads to the error accumulation and the sampling drift (Li et al., 2023a).

We point out that the exposure bias problem in diffusion models lacks in-depth exploration. For example, there is no proper metric to quantify the exposure bias and no explicit error analysis for it. To shed light on exposure bias, we conduct a systematical investigation by first modelling the sampling distribution with prediction error. Based on our analysis, we find that the practical sampling distribution has a larger variance than the ground truth distribution at every single step, demonstrating the

analytic difference between $\boldsymbol{x}_t$ in training and $\hat{\boldsymbol{x}}_t$ in sampling. Along with the sampling distribution analysis, we propose a metric $\delta_t$ to evaluate exposure bias by comparing the variance difference between training and sampling. Finally, we discuss potential solutions to exposure bias, and propose a simple yet effective *training-free and plug-in* method called Epsilon Scaling to alleviate this issue.

We test our approach on extensive diffusion frameworks using deterministic and stochastic sampling, and on conditional and unconditional generation tasks. Without affecting the recall and precision (Kynkäänniemi et al., 2019), our method yields dramatic Fréchet Inception Distance (FID) (Heusel et al., 2017) improvements. Also, we illustrate that Epsilon Scaling effectively reduces the exposure bias by moving the sampling trajectory towards the training trajectory. Overall, our contributions to diffusion models are:

- We investigate the exposure bias problem in depth and propose a metric for it.
- We suggest potential solutions to the exposure bias issue and propose a training-free, plug-in method (Epsilon Scaling) which significantly improves the sample quality.
- Our extensive experiments demonstrate the generality of Epsilon Scaling and its wide applicability to different diffusion architectures.

## 2 RELATED WORK

Diffusion models were introduced by Sohl-Dickstein et al. (2015) and later improved by Song & Ermon (2019), Ho et al. (2020) and Nichol & Dhariwal (2021). Song et al. (2021b) unify score-based models and Denoising Diffusion Probabilistic Models (DDPMs) via stochastic differential equations. Furthermore, Karras et al. (2022) disentangle the design space of diffusion models and introduce the EDM model to further boost the performance in image generation. With the advances in diffusion theory, conditional generation (Ho & Salimans, 2022; Choi et al., 2021) also flourishes in various scenarios, including text-to-image generation (Nichol et al., 2022; Ramesh et al., 2022; Rombach et al., 2022; Saharia et al., 2022), controllable image synthesis (Zhang & Agrawala, 2023; Li et al., 2023b; Zheng et al., 2023), as well as generating other modalities, for instance, audio (Chen et al., 2021; Kong et al., 2021), object shape (Zhou et al., 2021) and time series (Rasul et al., 2021). In the meantime, accelerating the time-consuming reverse diffusion sampling has been extensively investigated in many works (Song et al., 2021a; Lu et al., 2022; Liu et al., 2022). For example, distillation (Salimans & Ho, 2022), Restart sampler (Xu et al., 2023a) and fast ODE samplers (Zhao et al., 2023) have been proposed to speed up the sampling.

The exposure bias in diffusion models was first identified by Ning et al. (2023). They introduced a training regularisation term to simulate the sampling prediction errors from the Lipschitz continuity perspective. Additionally, Li et al. (2023a) alleviated exposure bias without retraining and their method involved the manipulation of the time step during the backward generation process. More recently, Li & van der Schaar (2023) estimated the upper bound of cumulative error and optimized it during training. However, the exposure bias in diffusion models still lacks illuminating research in terms of the explicit sampling distribution, metric and root cause, which is the objective of this paper. Besides, our solution to exposure bias is training-free and outperforms previous methods.

## 3 EXPOSURE BIAS IN DIFFUSION MODELS

### 3.1 SAMPLING DISTRIBUTION WITH PREDICTION ERROR

Given a sample $\boldsymbol{x}_0$ from the data distribution $q(\boldsymbol{x}_0)$ and a noise schedule $\beta_t$, DDPM (Ho et al., 2020) defines the forward perturbation as $q(\boldsymbol{x}_t|\boldsymbol{x}_{t-1}) = \mathcal{N}(\boldsymbol{x}_t; \sqrt{1-\beta_t}\boldsymbol{x}_{t-1}, \beta_t\boldsymbol{I})$. The Gaussian forward process allows us to directly sample $\boldsymbol{x}_t$ conditioned on the input $\boldsymbol{x}_0$:

$$q(\boldsymbol{x}_t|\boldsymbol{x}_0) = \mathcal{N}(\boldsymbol{x}_t; \sqrt{\bar{\alpha}_t}\boldsymbol{x}_0, (1-\bar{\alpha}_t)\boldsymbol{I}), \qquad \boldsymbol{x}_t = \sqrt{\bar{\alpha}_t}\boldsymbol{x}_0 + \sqrt{1-\bar{\alpha}_t}\boldsymbol{\epsilon}, \qquad (1)$$

The reverse diffusion is approximated by a neural network $p_{\boldsymbol{\theta}}(\boldsymbol{x}_{t-1}|\boldsymbol{x}_t) = \mathcal{N}(\boldsymbol{x}_{t-1}; \mu_{\boldsymbol{\theta}}(\boldsymbol{x}_t, t), \sigma_t\boldsymbol{I})$ and the optimisation objective is $D_{KL}(q(\boldsymbol{x}_{t-1}|\boldsymbol{x}_t, \boldsymbol{x}_0) \,\|\, p_{\boldsymbol{\theta}}(\boldsymbol{x}_{t-1}|\boldsymbol{x}_t)))$ in which the posterior $q(\boldsymbol{x}_{t-1}|\boldsymbol{x}_t, \boldsymbol{x}_0)$ is tractable when conditioned on $\boldsymbol{x}_0$ using Bayes theorem:

$$q(\boldsymbol{x}_{t-1}|\boldsymbol{x}_t, \boldsymbol{x}_0) = \mathcal{N}(\boldsymbol{x}_{t-1}; \tilde{\boldsymbol{\mu}}(\boldsymbol{x}_t, \boldsymbol{x}_0), \tilde{\beta}_t\boldsymbol{I}) \qquad (2)$$

$$\tilde{\boldsymbol{\mu}}(\boldsymbol{x}_t, \boldsymbol{x}_0) = \frac{\sqrt{\bar{\alpha}_{t-1}}\beta_t}{1 - \bar{\alpha}_t}\boldsymbol{x}_0 + \frac{\sqrt{\alpha_t}(1 - \bar{\alpha}_{t-1})}{1 - \bar{\alpha}_t}\boldsymbol{x}_t \qquad (3)$$

$$\tilde{\beta}_t = \frac{1 - \bar{\alpha}_{t-1}}{1 - \bar{\alpha}_t}\beta_t \qquad (4)$$

Regarding the parametrization of $\mu_{\boldsymbol{\theta}}(\boldsymbol{x}_t, t)$, Ho et al. (2020) found that using a neural network to predict $\boldsymbol{\epsilon}$ (Eq. 6) worked better than predicting $\boldsymbol{x}_0$ (Eq. 5) in practice:

$$\mu_{\boldsymbol{\theta}}(\boldsymbol{x}_t, t) = \frac{\sqrt{\bar{\alpha}_{t-1}}\beta_t}{1 - \bar{\alpha}_t}\boldsymbol{x}_{\boldsymbol{\theta}}(\boldsymbol{x}_t, t) + \frac{\sqrt{\alpha_t}(1 - \bar{\alpha}_{t-1})}{1 - \bar{\alpha}_t}\boldsymbol{x}_t \qquad (5)$$

$$= \frac{1}{\sqrt{\alpha_t}}(\boldsymbol{x}_t - \frac{\beta_t}{\sqrt{1 - \bar{\alpha}_t}}\boldsymbol{\epsilon}_{\boldsymbol{\theta}}(\boldsymbol{x}_t, t)), \qquad (6)$$

where $\boldsymbol{x}_{\boldsymbol{\theta}}(\boldsymbol{x}_t, t)$ denotes the denoising model which predicts $\boldsymbol{x}_0$ given $\boldsymbol{x}_t$. *For simplicity, we use $\boldsymbol{x}_{\boldsymbol{\theta}}^t$ as the short notation of $\boldsymbol{x}_{\boldsymbol{\theta}}(\boldsymbol{x}_t, t)$ in the rest of this paper.* Comparing Eq. 3 with Eq. 5, Song et al. (2021a) emphasise that the sampling distribution $p_{\boldsymbol{\theta}}(\boldsymbol{x}_{t-1}|\boldsymbol{x}_t)$ is in fact parameterised as $q(\boldsymbol{x}_{t-1}|\boldsymbol{x}_t, \boldsymbol{x}_{\boldsymbol{\theta}}^t)$ where $\boldsymbol{x}_{\boldsymbol{\theta}}^t$ means the predicted $\boldsymbol{x}_0$ given $\boldsymbol{x}_t$. Therefore, the practical sampling paradigm is that we first predict $\boldsymbol{\epsilon}$ using $\boldsymbol{\epsilon}_{\boldsymbol{\theta}}(\boldsymbol{x}_t, t)$. Then we derive the estimation $\boldsymbol{x}_{\boldsymbol{\theta}}^t$ for $\boldsymbol{x}_0$ using Eq. 1. Finally, based on the posterior $q(\boldsymbol{x}_{t-1}|\boldsymbol{x}_t, \boldsymbol{x}_0)$, $\boldsymbol{x}_{t-1}$ is generated using $q(\boldsymbol{x}_{t-1}|\boldsymbol{x}_t, \boldsymbol{x}_{\boldsymbol{\theta}}^t)$ by replacing $\boldsymbol{x}_0$ with $\boldsymbol{x}_{\boldsymbol{\theta}}^t$.

However, $q(\boldsymbol{x}_{t-1}|\boldsymbol{x}_t, \boldsymbol{x}_{\boldsymbol{\theta}}^t) = q(\boldsymbol{x}_{t-1}|\boldsymbol{x}_t, \boldsymbol{x}_0)$ holds only if $\boldsymbol{x}_{\boldsymbol{\theta}}^t = \boldsymbol{x}_0$, this requires the network to make no prediction error about $\boldsymbol{x}_0$ to ensure $q(\boldsymbol{x}_{t-1}|\boldsymbol{x}_t, \boldsymbol{x}_{\boldsymbol{\theta}}^t)$ share the same variance with $q(\boldsymbol{x}_{t-1}|\boldsymbol{x}_t, \boldsymbol{x}_0)$. However, $\boldsymbol{x}_{\boldsymbol{\theta}}^t - \boldsymbol{x}_0$ is practically non-zero and we claim that the prediction error of $\boldsymbol{x}_0$ needs to be considered to derive the real sampling distribution. Following Analytic-DPM (Bao et al., 2022), we model $\boldsymbol{x}_{\boldsymbol{\theta}}^t$ as $p_{\boldsymbol{\theta}}(\boldsymbol{x}_0|\boldsymbol{x}_t)$ and approximate it by a Gaussian distribution:

$$p_{\boldsymbol{\theta}}(\boldsymbol{x}_0|\boldsymbol{x}_t) = \mathcal{N}(\boldsymbol{x}_{\boldsymbol{\theta}}^t; \boldsymbol{x}_0, e_t^2\boldsymbol{I}), \qquad \boldsymbol{x}_{\boldsymbol{\theta}}^t = \boldsymbol{x}_0 + e_t\boldsymbol{\epsilon}_0 \quad (\boldsymbol{\epsilon}_0 \sim \mathcal{N}(\boldsymbol{0}, \boldsymbol{I})) \qquad (7)$$

Taking the prediction error into account, we now compute $q(\hat{\boldsymbol{x}}_t|\boldsymbol{x}_{t+1}, \boldsymbol{x}_{\boldsymbol{\theta}}^{t+1})$, which shares the same function format as $q(\boldsymbol{x}_{t-1}|\boldsymbol{x}_t, \boldsymbol{x}_{\boldsymbol{\theta}}^t)$, by substituting with the index $t+1$ and using $\hat{\boldsymbol{x}}_t$ to highlight that it is generated in the sampling stage. Based on Eq. 2, we know $q(\hat{\boldsymbol{x}}_t|\boldsymbol{x}_{t+1}, \boldsymbol{x}_{\boldsymbol{\theta}}^{t+1}) = \mathcal{N}(\hat{\boldsymbol{x}}_t; \mu_{\boldsymbol{\theta}}(\boldsymbol{x}_{t+1}, t+1), \tilde{\beta}_{t+1}\boldsymbol{I})$. Its mean and variance can be further derived according to Eq. 5 and Eq. 4, respectively. Thus, a sample from the distribution is $\hat{\boldsymbol{x}}_t = \mu_{\boldsymbol{\theta}}(\boldsymbol{x}_{t+1}, t+1) + \sqrt{\tilde{\beta}_{t+1}}\boldsymbol{\epsilon}_1$, namely:

$$\hat{\boldsymbol{x}}_t = \frac{\sqrt{\bar{\alpha}_t}\beta_{t+1}}{1 - \bar{\alpha}_{t+1}}\boldsymbol{x}_{\boldsymbol{\theta}}^{t+1} + \frac{\sqrt{\alpha_{t+1}}(1 - \bar{\alpha}_t)}{1 - \bar{\alpha}_{t+1}}\boldsymbol{x}_{t+1} + \sqrt{\tilde{\beta}_{t+1}}\boldsymbol{\epsilon}_1 \qquad (8)$$

Plugging Eq. 7 and Eq. 1 (using index $t+1$) into Eq. 8, we derive the final analytical form of $\hat{\boldsymbol{x}}_t$ (see Appendix A.1 for the full derivation):

$$\hat{\boldsymbol{x}}_t = \sqrt{\bar{\alpha}_t}\boldsymbol{x}_0 + \sqrt{1 - \bar{\alpha}_t + (\frac{\sqrt{\bar{\alpha}_t}\beta_{t+1}}{1 - \bar{\alpha}_{t+1}}e_{t+1})^2}\boldsymbol{\epsilon}_3 \qquad (9)$$

where, $\boldsymbol{\epsilon}_1, \boldsymbol{\epsilon}_3 \sim \mathcal{N}(\boldsymbol{0}, \boldsymbol{I})$. From Eq. 9, we can obtain the mean and variance of $q(\hat{\boldsymbol{x}}_t|\boldsymbol{x}_{t+1}, \boldsymbol{x}_{\boldsymbol{\theta}}^{t+1})$. *For simplicity, we denote $q(\hat{\boldsymbol{x}}_t|\boldsymbol{x}_{t+1}, \boldsymbol{x}_{\boldsymbol{\theta}}^{t+1})$ as $q_{\boldsymbol{\theta}}(\hat{\boldsymbol{x}}_t|\boldsymbol{x}_{t+1})$ from now on.* In Table 1, $q(\boldsymbol{x}_t|\boldsymbol{x}_0)$ shows the $\boldsymbol{x}_t$ seen by the network during training while $q_{\boldsymbol{\theta}}(\hat{\boldsymbol{x}}_t|\boldsymbol{x}_{t+1})$ indicates the $\hat{\boldsymbol{x}}_t$ exposed to the network during sampling. The method of solving $q_{\boldsymbol{\theta}}(\hat{\boldsymbol{x}}_t|\boldsymbol{x}_{t+1})$ can be generalised to multi-step sampling (detailed in Appendix A.2). In the same spirit of modelling $\boldsymbol{x}_{\boldsymbol{\theta}}^t$ as a Gaussian, we also derived the sampling distribution $q_{\boldsymbol{\theta}}(\hat{\boldsymbol{x}}_t|\boldsymbol{x}_{t+1})$ for DDIM (Song et al., 2021a) in Appendix A.3. Note that, one can also model $\boldsymbol{x}_{\boldsymbol{\theta}}^t$ as a Gaussian distribution whose mean is not $\boldsymbol{x}_0$, but it does not affect the variance gap presented in Table 1 and the explosion of variance error (Section 3.2).

Table 1: The distribution $q(\boldsymbol{x}_t|\boldsymbol{x}_0)$ during training and $q_{\boldsymbol{\theta}}(\hat{\boldsymbol{x}}_t|\boldsymbol{x}_{t+1})$ during DDPM sampling.

|  | Mean | Variance |
|---|---|---|
| $q(\boldsymbol{x}_t|\boldsymbol{x}_0)$ | $\sqrt{\bar{\alpha}_t}\boldsymbol{x}_0$ | $(1 - \bar{\alpha}_t)\boldsymbol{I}$ |
| $q_{\boldsymbol{\theta}}(\hat{\boldsymbol{x}}_t|\boldsymbol{x}_{t+1})$ | $\sqrt{\bar{\alpha}_t}\boldsymbol{x}_0$ | $(1 - \bar{\alpha}_t + (\frac{\sqrt{\bar{\alpha}_t}\beta_{t+1}}{1 - \bar{\alpha}_{t+1}}e_{t+1})^2)\boldsymbol{I}$ |

## 3.2 EXPOSURE BIAS DUE TO PREDICTION ERROR

It is clear from Table 1 that the variance of the sampling distribution $q_{\boldsymbol{\theta}}(\hat{\boldsymbol{x}}_t|\boldsymbol{x}_{t+1})$ is always larger than the variance of the training distribution $q(\boldsymbol{x}_t|\boldsymbol{x}_0)$ by the magnitude $(\frac{\sqrt{\bar{\alpha}_t}\beta_{t+1}}{1-\bar{\alpha}_{t+1}}e_{t+1})^2$. Note that, this variance gap between training and sampling is produced just in a single reverse diffusion step, given that the network $\boldsymbol{\epsilon}_{\boldsymbol{\theta}}(\cdot)$ can get access to the ground truth input $\boldsymbol{x}_{t+1}$. What makes the situation worse is that the error of single-step sampling accumulates in the multi-step sampling, resulting in an explosion of sampling variance error. On CIFAR-10 (Krizhevsky et al., 2009), we designed an experiment to statistically measure both the single-step variance error of $q_{\boldsymbol{\theta}}(\hat{\boldsymbol{x}}_t|\boldsymbol{x}_{t+1})$ and multi-step variance error of $q_{\boldsymbol{\theta}}(\hat{\boldsymbol{x}}_t|\boldsymbol{x}_T)$ using 20-step sampling (see Appendix A.4). The experimental results in Fig. 1 indicate that the closer to $t = 1$ (the end of sampling), the larger the variance error of multi-step sampling. The explosion of sampling variance error results in the sampling drift (exposure bias) problem and we attribute the prediction error $\boldsymbol{x}_{\boldsymbol{\theta}}^t - \boldsymbol{x}_0$ as the root cause of the exposure bias in diffusion models.

Intuitively, a possible solution to exposure bias is using a sampling noise variance which is smaller than $\tilde{\beta}_t$, to counteract the extra variance term $(\frac{\sqrt{\bar{\alpha}_t}\beta_{t+1}}{1-\bar{\alpha}_{t+1}}e_{t+1})^2$ caused by the prediction error $\boldsymbol{x}_{\boldsymbol{\theta}}^t - \boldsymbol{x}_0$. Unfortunately, $\tilde{\beta}_t$ is the lower bound of the sampling noise schedule $\dot{\beta}_t \in [\tilde{\beta}_t, \beta_t]$, where the lower bound and upper bound are the sampling variances given by $q(\boldsymbol{x}_0)$ being a delta function and isotropic Gaussian function, respectively (Nichol & Dhariwal, 2021). Therefore, we can conclude that the exposure bias problem can not be alleviated by manipulating the sampling noise schedule $\dot{\beta}_t$.

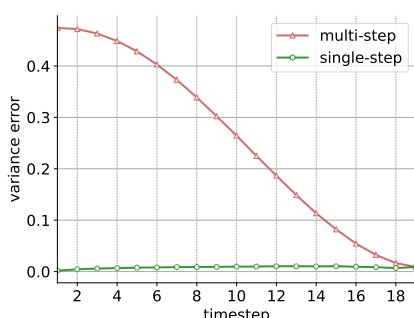

Figure 1: Variance error in single-step and multi-step samplings.

Interestingly, Bao et al. (2022) analytically provide the optimal sampling noise schedule $\beta_t^\star$ which is larger than the lower bound $\tilde{\beta}_t$. Based on what we discussed earlier, $\beta_t^\star$ would cause a more severe exposure bias issue than $\tilde{\beta}_t$. A strange phenomenon, but not explained by Bao et al. (2022) is that $\beta_t^\star$ leads to a worse FID than using $\tilde{\beta}_t$ under 1000 sampling steps. We believe the exposure bias is in the position to account for this phenomenon: under the long sampling, the negative impact of exposure bias exceeds the positive gain of the optimal variance $\beta_t^\star$.

## 3.3 METRIC FOR EXPOSURE BIAS

Although some literature has already discussed the exposure bias problem in diffusion models (Ning et al., 2023; Li et al., 2023a), we still lack a well-defined and straightforward metric for this concept. We propose to use the variance error of $q_{\boldsymbol{\theta}}(\hat{\boldsymbol{x}}_t|\boldsymbol{x}_T)$ to quantify the exposure bias at timestep $t$ under multi-step sampling. Specifically, our metric $\delta_t$ for exposure bias is defined as $\delta_t = (\sqrt{\beta_t'} - \sqrt{\bar{\beta}_t})^2$, where $\bar{\beta}_t = 1 - \bar{\alpha}_t$ denotes the variance of $q(\boldsymbol{x}_t|\boldsymbol{x}_0)$ during training and $\beta_t'$ presents the variance of $q_{\boldsymbol{\theta}}(\hat{\boldsymbol{x}}_t|\boldsymbol{x}_T)$ in the regular sampling process. The metric $\delta_t$ is inspired by the Fréchet distance (Dowson & Landau, 1982) between $q(\boldsymbol{x}_t|\boldsymbol{x}_0)$ and $q_{\boldsymbol{\theta}}(\hat{\boldsymbol{x}}_t|\boldsymbol{x}_T)$, which is $d^2 = N(\sqrt{\beta_t'} - \sqrt{\bar{\beta}_t})^2$ where $N$ is the dimensions of $\boldsymbol{x}_t$. In Appendix A.6, we empirically find that $\delta_t$ exhibits a strong correlation with FID given a trained model. Our method of measuring $\delta_t$ is described in Algorithm 3 (see Appendix A.5). The key step of Algorithm 3 is that we subtract the mean $\sqrt{\bar{\alpha}_{t-1}}\boldsymbol{x}_0$ and the remaining term $\hat{\boldsymbol{x}}_{t-1} - \sqrt{\bar{\alpha}_{t-1}}\boldsymbol{x}_0$ corresponds to the stochastic term of $q_{\boldsymbol{\theta}}(\hat{\boldsymbol{x}}_{t-1}|\boldsymbol{x}_T)$.

## 3.4 SOLUTION DISCUSSION

We now discuss possible solutions to the exposure bias issue of diffusion models based on the analysis throughout Section 3. Recall that the prediction error of $\boldsymbol{x}_{\boldsymbol{\theta}}^t - \boldsymbol{x}_0$ is the root cause of exposure bias. Thus, the most straightforward way of reducing exposure bias is learning an accurate $\boldsymbol{\epsilon}$ or score function (Song & Ermon, 2019) prediction network. For example, by delicately designing the network and hyper-parameters, EDM (Karras et al., 2022) significantly improves the FID from 3.01 to 2.51 on CIFAR-10. Secondly, we believe that data augmentation can reduce the risk of learning

inaccurate $\epsilon$ or score function for $\hat{x}_t$ by learning a denser vector field than vanilla diffusion models. For instance, Karras et al. (2022) has shown that the geometric augmentation (Karras et al., 2020) benefits the network training and sample quality. In the same spirit, Ning et al. (2023) augments each training sample $x_t$ by a Gaussian term and achieves substantial improvements in FID. Additionally, Xu et al. (2022) claim that the Poisson Flow framework is more resistant to prediction errors because high mass is allocated across a wide spectrum of the training sample. Our experiments verify the robustness of PGFM++ (Xu et al., 2023b) and show that our solution to exposure bias could further improve the generation quality of PGFM++ (more details in Appendix A.7).

It is worth pointing out that the above-mentioned methods require retraining the network and expensive parameter searching during the training. This naturally drives us to the question: *Can we alleviate the exposure bias in the sampling stage, without any retraining*?

## 4 METHOD

### 4.1 EPSILON SCALING

In Section 3.2, we have concluded that the exposure bias issue can not be solved by reducing the sampling noise variance, thus another direction to be explored in the sampling phase is the prediction of the network $\epsilon_\theta(\cdot)$. Since we already know from Table 1 that $x_t$ inputted to the network $\epsilon_\theta(\cdot)$ in training differs from $\hat{x}_t$ fed into the network $\epsilon_\theta(\cdot)$ in sampling, we are interested in understanding the difference in the output of $\epsilon_\theta(\cdot)$ between training and inference.

For simplicity, we denote the output of $\epsilon_\theta(\cdot)$ as $\epsilon_\theta^t$ in training and as $\epsilon_\theta^s$ in sampling. Although the ground truth of $\epsilon_\theta^s$ is not accessible during inference, we are still able to speculate the behaviour of $\epsilon_\theta^s$ from the L2-norm perspective. In Fig. 2, we plot the L2-norm of $\epsilon_\theta^t$ and $\epsilon_\theta^s$ at each timestep. In detail, given a trained, frozen model and ground truth $x_t$, $\epsilon_\theta^t$ is collected by $\epsilon_\theta^t = \epsilon_\theta(x_t, t)$. In this way, we simulate the training stage and analyse its $\epsilon$ prediction. In contrast, $\epsilon_\theta^s$ is gathered in the real sampling process, namely $\epsilon_\theta^s = \epsilon_\theta(\hat{x}_t, t)$. It is clear from Fig. 2 that the L2-norm of $\epsilon_\theta^s$ is always larger than that of $\epsilon_\theta^t$. Since $\hat{x}_t$ lies around $x_t$ with a larger variance (Section 3.1), we can know the network learns an inaccurate vector field $\epsilon_\theta(\hat{x}_t, t)$ for each $\hat{x}_t$ in the vicinity of $x_t$ with the vector length longer than that of $\epsilon_\theta(x_t, t)$.

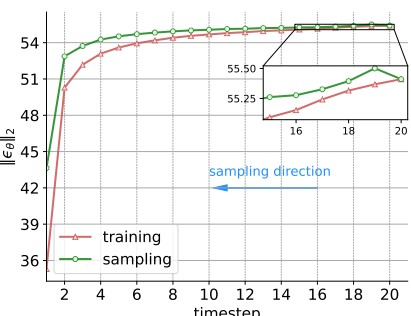

Figure 2: Expectation of $\|\epsilon_\theta(\cdot)\|_2$ during training and 20-step sampling on CIFAR-10. We report the L2-norm using 50k samples at each timestep.

One can infer that the prediction $\epsilon_\theta^s$ could be improved if we can move the input $(\hat{x}_t, t)$ from the inaccurate vector field (green curve in Fig. 2) towards the reliable vector field (red curve in Fig. 2). To this end, we propose to scale down $\epsilon_\theta^s$ by a factor $\lambda(t)$ at sampling timestep $t$. Our solution is based on the observation: $\epsilon_\theta^t$ and $\epsilon_\theta^s$ share the same input $x_T \sim \mathcal{N}(0, I)$ at timestep $t = T$, but from timestep $T - 1$, $\hat{x}_t$ (input of $\epsilon_\theta^s$) starts to diverge from $x_t$ (input of $\epsilon_\theta^t$) due to the $\epsilon_\theta(\cdot)$ error made at previous timestep. This iterative process continues along the sampling chain and results in exposure bias. Therefore, we can push $\hat{x}_t$ closer to $x_t$ by scaling down the over-predicted magnitude of $\epsilon_\theta^s$. Compared with the regular sampling (Eq. 6), our sampling method only differs in the $\lambda_t$ term and is expressed as $\mu_\theta(x_t, t) = \frac{1}{\sqrt{\alpha_t}}(x_t - \frac{\beta_t}{\sqrt{1-\bar{\alpha}_t}} \frac{\epsilon_\theta(x_t, t)}{\lambda_t})$. Note that, our Epsilon Scaling serving as a plug-in method adds no computational load to the original sampling of diffusion models.

### 4.2 THE DESIGN OF SCALING SCHEDULE

Similar to the cumulative error analysed in Li & van der Schaar (2023), we emphasise that the L2-norm quotient $\|\epsilon_\theta^s\|_2 / \|\epsilon_\theta^t\|_2$, denoted as $\Delta N(t)$, reflects the accumulated effect of the prediction errors made at ancestral steps $T, T-1, ..., t+1$. Suppose the L2-norm of $\epsilon_\theta^t$ to be scaled at timestep $t$ is $\lambda_t$, we have:

$$\Delta N(t) - 1 \approx \int_t^T (\lambda_T - 1)dt + \int_t^{T-1} (\lambda_{T-1} - 1)dt + ... + \int_t^{t+1} (\lambda_{t+1} - 1)dt \quad (10)$$

given that the error propagates linearly over the sampling chain. The first term on the right side of Eq. 10 corresponds to the error made at timestep $T$ (the start of sampling) and propagated from $T$ to $t$. Thus, after measuring $\Delta N(t)$, one can derive the scaling schedule $\lambda(t)$ by solving Eq. 10.

As shown by Nichol & Dhariwal (2021) and Benny & Wolf (2022), the $\epsilon_{\theta}(\cdot)$ predictions near $t = 0$ are very bad, with the loss larger than other timesteps by several orders of magnitude. Thereby, we can ignore the area close to $t = 0$ to fit $\Delta N(t)$, because scaling a problematic $\epsilon_{\theta}(\cdot)$ does not lead to a better prediction. We plot the $\Delta N(t)$ curve in the cases of 20-step and 50-step sampling on CIFAR-10 in Fig. 3 where $\Delta N(t)$ can be fitted by a quadratic function in the interval $t \sim (5, T)$. Thus, the solution to Eq. 10 is a linear function $\lambda(t) = kt + b$ where $k, b$ are constants. Ideally, one should always first do simulations to measure $\left\|\epsilon_{\theta}^{s}\right\|_{2}$ and $\left\|\epsilon_{\theta}^{t}\right\|_{2}$, then solve out $\lambda(t)$ based on $\Delta N(t)$. However, we propose to search for $k, b$ because we find that the parameters do not change significantly in different networks. An added benefit of this proposal is that our approach becomes simulation-free. Moreover, in Section 5.1, we will see that $k$ would decay to 0 around 50-step sampling. Given this fact, we decide a uniform $\lambda(t) = b$ for most experiments because of its effortless parameter searching and near-optimal performance.

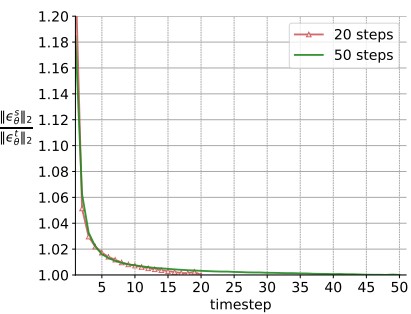

Figure 3: $\Delta N(t)$ at each timestep.

## 5  RESULTS

In this section, we evaluate the performance of Epsilon Scaling using FID (Heusel et al., 2017). To demonstrate that Epsilon Scaling is a generic solution to exposure bias, we test the approach on various diffusion frameworks, samplers and conditional settings. Following the fast sampling paradigm (Karras et al., 2022) in the diffusion community, we focus on $T' \leqslant 100$ sampling steps in this section and leave the FID results of $T' > 100$ in Appendix A.8. Our FID computation is consistent with (Dhariwal & Nichol, 2021) for equal comparison. All FIDs are reported using 50k generated samples and the full training set as the reference batch. Lastly, Epsilon Scaling does not affect the precision and recall, and we report these results in Appendix A.9.

### 5.1  MAIN RESULTS ON ADM

Since Epsilon Scaling is a training-free method, we utilise the pre-trained ADM model as the baseline and compare it against our ADM-ES (ADM with Epsilon Scaling) on datasets CIFAR-10 (Krizhevsky et al., 2009), LSUN tower (Yu et al., 2015) and FFHQ (Karras et al., 2019) for unconditional generation and on datasets ImageNet 64×64 and ImageNet 128×128 (Chrabaszcz et al., 2017) for class-conditional generation. We employ the respacing sampling technique (Nichol & Dhariwal, 2021) to enable fast stochastic sampling.

Table 2: FID on ADM baseline. We compare ADM with our ADM-ES (uniform $\lambda(t)$) and ADM-ES* (linear $\lambda(t)$). ImageNet 64×64 results are reported without classifier guidance and ImageNet 128×128 is under classifier guidance with scale=0.5

| $T'$ | Model | Unconditional | | | Conditional | |
|---|---|---|---|---|---|---|
| | | CIFAR-10 32×32 | LSUN 64×64 | FFHQ 128×128 | ImageNet 64×64 | ImageNet 128×128 |
| 100 | ADM | 3.37 | 3.59 | 14.52 | 2.71 | 3.55 |
| | ADM-ES | **2.17** | **2.91** | **6.77** | **2.39** | **3.37** |
| 50 | ADM | 4.43 | 7.28 | 26.15 | 3.75 | 5.15 |
| | ADM-ES | **2.49** | **3.68** | **9.50** | **3.07** | **4.33** |
| 20 | ADM | 10.36 | 23.92 | 59.35 | 10.96 | 12.48 |
| | ADM-ES | 5.15 | 8.22 | 26.14 | 7.52 | 9.95 |
| | ADM-ES* | **4.31** | **7.60** | **24.83** | **7.37** | **9.86** |

Table 3: We compare ADM-ES with recent stochastic diffusion (SDE) samplers regarding FID. We report their best FID achieved under $T'$ sampling steps.

| Model | $T'$ | Unconditional |
|---|---|---|
| | | CIFAR-10 32×32 |
| EDM (VP) (Karras et al., 2022) | 511 | 2.27 |
| EDM (VE) (Karras et al., 2022) | 2047 | 2.23 |
| Improved SDE (Karras et al., 2022) | 1023 | 2.35 |
| Restart (VP) (Xu et al., 2023a) | 115 | 2.21 |
| SA-Solver (Xue et al., 2023) | 95 | 2.63 |
| ADM-IP (Ning et al., 2023) | 100 | 2.38 |
| ADM-ES (ours) | **50** | 2.49 |
| ADM-ES (ours) | 100 | **2.17** |

Table 2 shows that independent of datasets and the number of sampling steps $T'$, our ADM-ES outperforms ADM by a large margin in terms of FID. For instance, on FFHQ 128×128, ADM-ES exhibits less than half the FID of ADM, with 7.75, 16.65 and 34.52 FID improvement under 100, 50 and 20 sampling steps, respectively. Moreover, when compared with the previous best stochastic samplers, ADM-ES outperforms EDM (Karras et al., 2022), Improved SDE (Karras et al., 2022), Restart Sampler (Xu et al., 2023a) and SA-Solver (Xue et al., 2023), exhibiting state-of-the-art stochastic sampler (SDE solver). For example, ADM-ES not only achieves a better FID (2.17) than EDM and Improved SDE, but also accelerates the sampling speed by 5-fold to 20-fold (see Table 3). Even under 50-step sampling, Epsilon Scaling surpasses SA-Solver and obtains competitive FID against other samplers.

Note that, ADM-ES uses uniform schedule $\lambda(t) = b$ and ADM-ES$^*$ applies the linear schedule $\lambda(t) = kt + b$ in Table 2. We find that the slope $k$ is approaching 0 as the sampling step $T'$ increases. Therefore, we suggest a uniform schedule $\lambda(t)$ for practical consideration. We present the complete parameters $k, b$ used in all experiments and the details on the search of $k, b$ in Appendix A.10. Overall, searching for the optimal uniform $\lambda(t)$ is effortless and takes 6 to 10 trials. In Appendix A.11, we also demonstrate that the FID gain can be achieved within a wide range of $\lambda(t)$, which indicates the insensitivity of $\lambda(t)$.

## 5.2 EPSILON SCALING ALLEVIATES EXPOSURE BIAS

Apart from the FID improvements, we now show the exposure bias alleviated by our method using the proposed metric $\delta_t$ and we also demonstrate the sampling trajectory corrected by Epsilon Scaling. Using Algorithm 3, we measure $\delta_t$ on the dataset CIFAR-10 under 20-step sampling for ADM and ADM-ES models. Fig. 4 shows that ADM-ES obtains a lower $\delta_t$ at the end of sampling $t = 1$ than the baseline ADM, exhibiting a smaller variance error and sampling drift (see Appendix A.12 for results on other datasets).

Based on Fig. 2, we apply the same method to measure the L2-norm of $\boldsymbol{\epsilon_\theta}(\cdot)$ in the sampling phase with Epsilon Scaling. Fig. 5 indicates that our method explicitly moves the original sampling trajectory closer to the vector field learned in the training phase given the condition that the $\|\boldsymbol{\epsilon_\theta}(\boldsymbol{x}_t)\|_2$ is locally monotonic around $\boldsymbol{x}_t$. This condition is satisfied in denoising networks (Goodfellow et al., 2016; Song & Ermon, 2019) because of the monotonic score vectors around the local maximal probability density. We emphasise that Epsilon Scaling corrects the magnitude error of $\boldsymbol{\epsilon_\theta}(\cdot)$, but not the direction error. Thus we can not completely eliminate the exposure bias to achieve $\delta_t = 0$ or push the sampling trajectory to the exact training vector field.

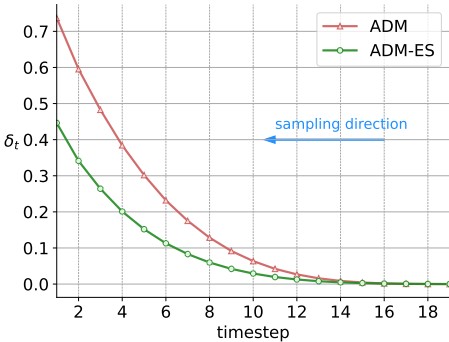
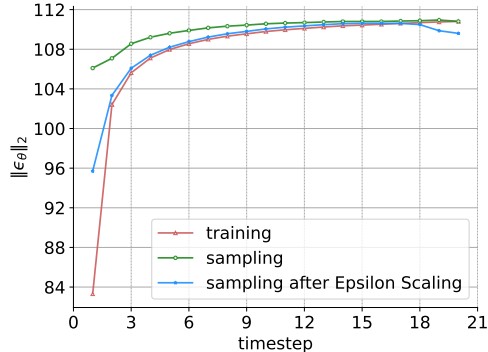

Figure 4: Exposure bias measured by $\delta_t$ on LSUN 64×64. Epsilon Scaling achieves a smaller $\delta_t$ at the end of sampling ($t = 1$)

Figure 5: $\|\boldsymbol{\epsilon_\theta}(\cdot)\|_2$ on LSUN 64×64. After applying Epsilon Scaling, the sampling $\|\boldsymbol{\epsilon_\theta}\|_2$ (blue) gets closer to the training $\|\boldsymbol{\epsilon_\theta}\|_2$ (red).

## 5.3 RESULTS ON DDIM/DDPM

To show the generality of our proposed method, we conduct experiments on DDIM/DDPM framework across CIFAR-10 and CelebA 64×64 datasets (Liu et al., 2015). The results are detailed in Table 4, wherein the designations $\eta = 0$ and $\eta = 1$ correspond to DDIM and DDPM samplers, respectively. The findings in Table 3 illustrate that our method can further boost the performance

of both DDIM and DDPM samplers on the CIFAR-10 and CelebA datasets. Specifically, our proposed Epsilon Scaling technique improves the performance of DDPM sampler on CelebA dataset by $47.7\%$, $63.1\%$, $60.7\%$ with 20, 50, and 100 sampling steps, respectively. Similar performance improvement can also be observed on CIFAR-10 dataset. We also notice that our method brings less performance improvement for DDIM sampler. This could arise from the FID advantage of deterministic sampling under a short sampling chain and the noise term in DDPM sampler can actively correct for errors made in earlier sampling steps Karras et al. (2022).

Table 4: FID on DDIM baseline for unconditional generations.

| $T'$ | Model | CIFAR-10 $32\times32$ | | CelebA $64\times64$ | |
|---|---|---|---|---|---|
| | | $\eta = 0$ | $\eta = 1$ | $\eta = 0$ | $\eta = 1$ |
| 100 | DDIM | 4.06 | 6.73 | 5.67 | 11.33 |
| | DDIM-ES (ours) | **3.38** | **4.01** | **5.05** | **4.45** |
| 50 | DDIM | 4.82 | 10.29 | 6.88 | 15.09 |
| | DDIM-ES | **4.17** | **4.57** | **6.20** | **5.57** |
| 20 | DDIM | 8.21 | 20.15 | 10.43 | 22.61 |
| | DDIM-ES | **6.54** | **7.78** | **10.38** | **11.83** |

Table 5: FID on EDM baseline and CIFAR-10 dateset (FID of EDM is reproduced).

| $T'$ | Model | Unconditional | | Conditional | |
|---|---|---|---|---|---|
| | | Heun | Euler | Heun | Euler |
| 35 | EDM | 1.97 | 3.81 | 1.82 | 3.74 |
| | EDM-ES (ours) | **1.95** | **2.80** | **1.80** | **2.59** |
| 21 | EDM | 2.33 | 6.29 | 2.17 | 5.91 |
| | EDM-ES | **2.24** | **4.32** | **2.08** | **3.74** |
| 13 | EDM | 7.16 | 12.28 | 6.69 | 10.66 |
| | EDM-ES | **6.54** | **8.39** | **6.16** | **6.59** |

## 5.4 RESULTS ON EDM

We test the effectiveness of Epsilon Scaling on EDM (Karras et al., 2022) because it achieves state-of-the-art image generation under a few sampling steps and provides a unified framework for diffusion models. Since the main advantage of EDM is its Ordinary Differential Equation (ODE) solver, we evaluate our Epsilon Scaling using their Heun $2^{nd}$ order ODE solver (Ascher & Petzold, 1998) and Euler $1^{st}$ order ODE solver, respectively. Although the network output of EDM is not $\epsilon$, we still can extract the signal $\epsilon$ at each sampling step and then apply Epsilon Scaling on $\epsilon$.

The experiments are implemented on CIFAR-10 dataset and we report the FID results in Table 5 using VP framework. The sampling step $T'$ in Table 5 is equivalent to the Neural Function Evaluations (NFE) used in EDM paper. Similar to the results on ADM and DDIM, Epsilon Scaling gains consistent FID improvement on EDM baseline regardless of the conditional settings and the ODE solver types. For instance, EDM-ES improves the FID from 3.81 to 2.80 and from 3.74 to 2.59 in the unconditional and conditional groups using the 35-step Euler sampler.

An interesting phenomenon in Table 5 is that the FID gain of Epsilon Scaling in the Euler sampler group is larger than that in the Heun sampler group. We believe there are two factors accounting for this phenomenon. On the one hand, higher-order ODE solvers (for example, Heun solvers) introduce less truncation error than Euler $1^{st}$ order solvers. On the other hand, the correction steps in the Heun solver reduce the exposure bias by pulling the drifted sampling trajectory back to the accurate vector field. We illustrate these two factors through Fig. 6 which is plotted using the same method of Fig. 2. It is apparent from Fig. 6 that the Heun sampler exhibits a smaller gap between the training trajectory and sampling trajectory when compared with the Euler sampler. This corresponds to the truncation error factor in these two ODE solvers. Furthermore, in the Heun $2^{nd}$ ODE sampler, the prediction error (cause of exposure bias) made in each Euler step is corrected in the subsequent Correction step (Fig. 6(b)), resulting in a reduced exposure bias. This exposure bias perspective explains the superiority of the Heun solver in diffusion models beyond the truncation error viewpoint.

## 5.5 RESULTS ON LDM

To further verify the generality of Epsilon Scaling, we adopt Latent Diffusion Model (LDM) as the base model which introduces an Autoeoconder and performs the diffusion process in the latent space (Rombach et al., 2022). We test the performance of Epsilon Scaling (LDM-ES) on FFHQ $256\times256$ and CelebA-HQ $256\times256$ datasets using $T'$ steps DDPM sampler. It is clear from Table 6 that Epsilon Scaling gains substantial FID improvements on the two high-resolution datasets, where LDM-ES achieves 15.68 FID under $T' = 20$ on CelebA-HQ, almost half that of LDM.

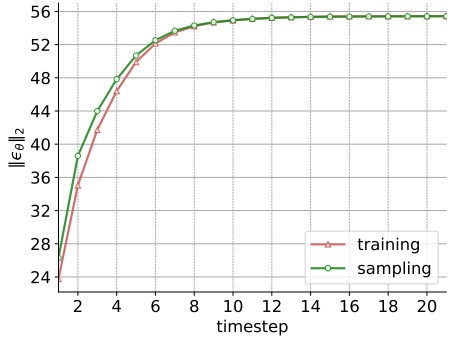
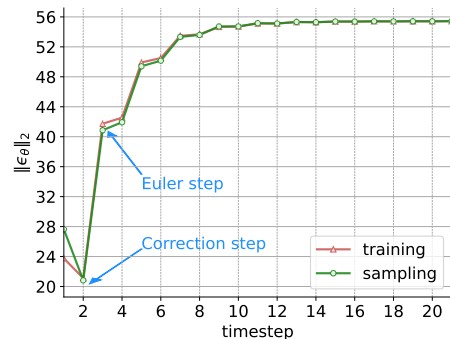

(a) EDM: Euler $1^{st}$ order sampler

(b) EDM: Heun $2^{nd}$ order sampler

Figure 6: $\|\boldsymbol{\epsilon}_{\boldsymbol{\theta}}(\cdot)\|_2$ during training and sampling on CIFAR-10. We use 21-step sampling and report the L2-norm using 50k samples at each timestep. The sampling is from right to left in the figures.

Epsilon Scaling also yields better FID under 50 and 100 sampling steps on CelebA-HQ with 7.36 FID at $T' = 100$. Similar FID improvements are obtained on FFHQ dataset over different $T'$.

Finally, Epsilon Scaling is also effective on DiT (Peebles & Xie, 2023) which applies the ViT (Dosovitskiy et al., 2020) diffusion backbone in LDM. Please refer to Appendix A.13 for the FID results on DiT baseline.

Table 6: FID on LDM baseline using DDPM unconditional sampling.

| $T'$ | Model | FFHQ 256×256 | CelebA-HQ 256×256 |
|---|---|---|---|
| 100 | LDM | 10.90 | 9.31 |
| | LDM-ES (ours) | **9.83** | **7.36** |
| 50 | LDM | 14.34 | 13.95 |
| | LDM-ES | **11.57** | **9.16** |
| 20 | LDM | 33.13 | 29.62 |
| | LDM-ES | **20.91** | **15.68** |

### 5.6 QUALITATIVE COMPARISON

In order to visually show the effect of Epsilon Scaling on image synthesis, we set the same random seed for the base model and our Epsilon Scaling model in the sampling phase to ensure a similar trajectory for both models. Fig. 7 displays the generated samples using 100 steps on FFHQ 128×128 dataset. It is clear that ADM-ES effectively refines the sample issues of ADM, including overexposure, underexposure, coarse background and detail defects from left to right in Fig. 7 (see Appendix A.14 for more qualitative comparisons). Besides, the qualitative com-

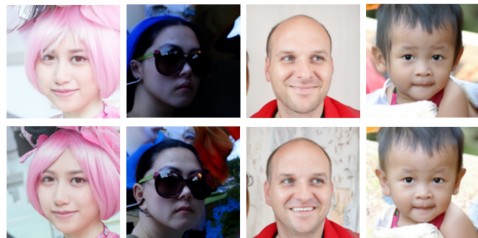

Figure 7: Qualitative comparison between ADM (first row) and ADM-ES (second row).

parison also empirically confirms that Epsilon Scaling guides the sampling trajectory of the base model to an adjacent but better probability path because both models reach the same or similar modes given the common starting point $\boldsymbol{x}_T$ and the same random seed at each sampling step.

## 6 CONCLUSIONS

In this paper, we elucidate the exposure bias issue in diffusion models by analytically showing the difference between the training distribution and sampling distribution. Moreover, we propose a training-free method to refine the deficient sampling trajectory by explicitly scaling the prediction vector. Through extensive experiments, we demonstrate that Epsilon Scaling is a generic solution to exposure bias and its simplicity enables a wide range of diffusion applications.

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

# A  APPENDIX

## A.1  DERIVATION OF $q_{\boldsymbol{\theta}}(\hat{\boldsymbol{x}}_t|\boldsymbol{x}_{t+1})$ FOR DDPM

We show the full derivation of Eq. 3 below. From Eq. 11 to Eq. 12, we plug in $\boldsymbol{x}_{\boldsymbol{\theta}}^{t+1} = \boldsymbol{x}_0 + e_{t+1}\boldsymbol{\epsilon}_0$ (Eq. 7) and $\boldsymbol{x}_{t+1} = \sqrt{\bar{\alpha}_{t+1}}\boldsymbol{x}_0 + \sqrt{1 - \bar{\alpha}_{t+1}}\boldsymbol{\epsilon}$ (Eq. 1), thus a sample from $q_{\boldsymbol{\theta}}(\hat{\boldsymbol{x}}_t|\boldsymbol{x}_{t+1})$ is:

$$
\begin{aligned}
\hat{\boldsymbol{x}}_t &= \mu_{\boldsymbol{\theta}}(\boldsymbol{x}_{t+1}, t+1) + \sqrt{\tilde{\beta}_{t+1}}\boldsymbol{\epsilon}_1 \\
&= \frac{\sqrt{\bar{\alpha}_t}\beta_{t+1}}{1-\bar{\alpha}_{t+1}}\boldsymbol{x}_{\boldsymbol{\theta}}^{t+1} + \frac{\sqrt{\alpha_{t+1}}(1-\bar{\alpha}_t)}{1-\bar{\alpha}_{t+1}}\boldsymbol{x}_{t+1} + \sqrt{\tilde{\beta}_{t+1}}\boldsymbol{\epsilon}_1 \quad &(11) \\
&= \frac{\sqrt{\bar{\alpha}_t}\beta_{t+1}}{1-\bar{\alpha}_{t+1}}(\boldsymbol{x}_0 + e_{t+1}\boldsymbol{\epsilon}_0) + \frac{\sqrt{\alpha_{t+1}}(1-\bar{\alpha}_t)}{1-\bar{\alpha}_{t+1}}(\sqrt{\bar{\alpha}_{t+1}}\boldsymbol{x}_0 + \sqrt{1-\bar{\alpha}_{t+1}}\boldsymbol{\epsilon}) + \sqrt{\tilde{\beta}_{t+1}}\boldsymbol{\epsilon}_1 \quad &(12) \\
&= \frac{\sqrt{\bar{\alpha}_t}\beta_{t+1}}{1-\bar{\alpha}_{t+1}}\boldsymbol{x}_0 + \frac{\sqrt{\alpha_{t+1}}(1-\bar{\alpha}_t)}{1-\bar{\alpha}_{t+1}}\sqrt{\bar{\alpha}_{t+1}}\boldsymbol{x}_0 + \frac{\sqrt{\bar{\alpha}_t}\beta_{t+1}}{1-\bar{\alpha}_{t+1}}e_{t+1}\boldsymbol{\epsilon}_0 + \frac{\sqrt{\alpha_{t+1}}(1-\bar{\alpha}_t)}{1-\bar{\alpha}_{t+1}}\sqrt{1-\bar{\alpha}_{t+1}}\boldsymbol{\epsilon} + \sqrt{\tilde{\beta}_{t+1}}\boldsymbol{\epsilon}_1 \\
&= \frac{\sqrt{\bar{\alpha}_t}\beta_{t+1} + \sqrt{\alpha_{t+1}}(1-\bar{\alpha}_t)\sqrt{\bar{\alpha}_{t+1}}}{1-\bar{\alpha}_{t+1}}\boldsymbol{x}_0 + \frac{\sqrt{\bar{\alpha}_t}\beta_{t+1}}{1-\bar{\alpha}_{t+1}}e_{t+1}\boldsymbol{\epsilon}_0 + \frac{\sqrt{\alpha_{t+1}}(1-\bar{\alpha}_t)}{1-\bar{\alpha}_{t+1}}\sqrt{1-\bar{\alpha}_{t+1}}\boldsymbol{\epsilon} + \sqrt{\tilde{\beta}_{t+1}}\boldsymbol{\epsilon}_1 \\
&= \frac{\sqrt{\bar{\alpha}_t}(1-\alpha_{t+1}) + \sqrt{\alpha_{t+1}}(1-\bar{\alpha}_t)\sqrt{\bar{\alpha}_{t+1}}}{1-\bar{\alpha}_{t+1}}\boldsymbol{x}_0 + \frac{\sqrt{\bar{\alpha}_t}\beta_{t+1}}{1-\bar{\alpha}_{t+1}}e_{t+1}\boldsymbol{\epsilon}_0 + \frac{\sqrt{\alpha_{t+1}}(1-\bar{\alpha}_t)}{1-\bar{\alpha}_{t+1}}\sqrt{1-\bar{\alpha}_{t+1}}\boldsymbol{\epsilon} + \sqrt{\tilde{\beta}_{t+1}}\boldsymbol{\epsilon}_1 \\
&= \frac{\sqrt{\bar{\alpha}_t}(1-\alpha_{t+1}) + \alpha_{t+1}(1-\bar{\alpha}_t)\sqrt{\bar{\alpha}_t}}{1-\bar{\alpha}_{t+1}}\boldsymbol{x}_0 + \frac{\sqrt{\bar{\alpha}_t}\beta_{t+1}}{1-\bar{\alpha}_{t+1}}e_{t+1}\boldsymbol{\epsilon}_0 + \frac{\sqrt{\alpha_{t+1}}(1-\bar{\alpha}_t)}{1-\bar{\alpha}_{t+1}}\sqrt{1-\bar{\alpha}_{t+1}}\boldsymbol{\epsilon} + \sqrt{\tilde{\beta}_{t+1}}\boldsymbol{\epsilon}_1 \\
&= \frac{\sqrt{\bar{\alpha}_t}(1-\alpha_{t+1}+\alpha_{t+1}-\bar{\alpha}_{t+1})}{1-\bar{\alpha}_{t+1}}\boldsymbol{x}_0 + \frac{\sqrt{\bar{\alpha}_t}\beta_{t+1}}{1-\bar{\alpha}_{t+1}}e_{t+1}\boldsymbol{\epsilon}_0 + \frac{\sqrt{\alpha_{t+1}}(1-\bar{\alpha}_t)}{1-\bar{\alpha}_{t+1}}\sqrt{1-\bar{\alpha}_{t+1}}\boldsymbol{\epsilon} + \sqrt{\tilde{\beta}_{t+1}}\boldsymbol{\epsilon}_1 \\
&= \sqrt{\bar{\alpha}_t}\boldsymbol{x}_0 + \frac{\sqrt{\bar{\alpha}_t}\beta_{t+1}}{1-\bar{\alpha}_{t+1}}e_{t+1}\boldsymbol{\epsilon}_0 + \frac{\sqrt{\alpha_{t+1}}(1-\bar{\alpha}_t)}{1-\bar{\alpha}_{t+1}}\sqrt{1-\bar{\alpha}_{t+1}}\boldsymbol{\epsilon} + \sqrt{\tilde{\beta}_{t+1}}\boldsymbol{\epsilon}_1 \quad &(13)
\end{aligned}
$$

where, $\boldsymbol{\epsilon}_0, \boldsymbol{\epsilon}, \boldsymbol{\epsilon}_1 \sim \mathcal{N}(\boldsymbol{0}, \boldsymbol{I})$. From Eq. 13, we know that the mean of $q_{\boldsymbol{\theta}}(\hat{\boldsymbol{x}}_t|\boldsymbol{x}_{t+1})$ is $\sqrt{\bar{\alpha}_t}\boldsymbol{x}_0$. We now focus on the variance by looking at $\frac{\sqrt{\bar{\alpha}_t}\beta_{t+1}}{1-\bar{\alpha}_{t+1}}e_{t+1}\boldsymbol{\epsilon}_0 + \frac{\sqrt{\alpha_{t+1}}(1-\bar{\alpha}_t)}{1-\bar{\alpha}_{t+1}}\sqrt{1-\bar{\alpha}_{t+1}}\boldsymbol{\epsilon} + \sqrt{\tilde{\beta}_{t+1}}\boldsymbol{\epsilon}_1$:

$$
\begin{aligned}
Var(\hat{\boldsymbol{x}}_t) &= (\frac{\sqrt{\bar{\alpha}_t}\beta_{t+1}}{1-\bar{\alpha}_{t+1}}e_{t+1})^2 + (\frac{\sqrt{\alpha_{t+1}}(1-\bar{\alpha}_t)}{1-\bar{\alpha}_{t+1}}\sqrt{1-\bar{\alpha}_{t+1}})^2 + \tilde{\beta}_{t+1} \quad &(14) \\
&= (\frac{\sqrt{\bar{\alpha}_t}\beta_{t+1}}{1-\bar{\alpha}_{t+1}}e_{t+1})^2 + (\frac{\sqrt{\alpha_{t+1}}(1-\bar{\alpha}_t)}{1-\bar{\alpha}_{t+1}}\sqrt{1-\bar{\alpha}_{t+1}})^2 + \frac{(1-\bar{\alpha}_t)(1-\alpha_{t+1})}{1-\bar{\alpha}_{t+1}} \\
&= (\frac{\sqrt{\bar{\alpha}_t}\beta_{t+1}}{1-\bar{\alpha}_{t+1}}e_{t+1})^2 + \frac{\alpha_{t+1}(1-\bar{\alpha}_t)^2}{1-\bar{\alpha}_{t+1}} + \frac{(1-\bar{\alpha}_t)(1-\alpha_{t+1})}{1-\bar{\alpha}_{t+1}} \\
&= (\frac{\sqrt{\bar{\alpha}_t}\beta_{t+1}}{1-\bar{\alpha}_{t+1}}e_{t+1})^2 + \frac{\alpha_{t+1}(1-\bar{\alpha}_t)^2 + (1-\bar{\alpha}_t)(1-\alpha_{t+1})}{1-\bar{\alpha}_{t+1}} \\
&= (\frac{\sqrt{\bar{\alpha}_t}\beta_{t+1}}{1-\bar{\alpha}_{t+1}}e_{t+1})^2 + \frac{(1-\bar{\alpha}_t)[\alpha_{t+1}(1-\bar{\alpha}_t) + (1-\alpha_{t+1})]}{1-\bar{\alpha}_{t+1}} \\
&= (\frac{\sqrt{\bar{\alpha}_t}\beta_{t+1}}{1-\bar{\alpha}_{t+1}}e_{t+1})^2 + \frac{(1-\bar{\alpha}_t)[\alpha_{t+1} - \bar{\alpha}_{t+1} + 1 - \alpha_{t+1}]}{1-\bar{\alpha}_{t+1}} \\
&= (\frac{\sqrt{\bar{\alpha}_t}\beta_{t+1}}{1-\bar{\alpha}_{t+1}}e_{t+1})^2 + \frac{(1-\bar{\alpha}_t)[1-\bar{\alpha}_{t+1}]}{1-\bar{\alpha}_{t+1}} \\
&= (\frac{\sqrt{\bar{\alpha}_t}\beta_{t+1}}{1-\bar{\alpha}_{t+1}}e_{t+1})^2 + 1 - \bar{\alpha}_t \quad &(15)
\end{aligned}
$$

## A.2 DERIVATION OF $q_{\boldsymbol{\theta}}(\hat{\boldsymbol{x}}_{t-1}|\boldsymbol{x}_{t+1})$ AND MORE FOR DDPM

Since $q_{\boldsymbol{\theta}}(\hat{\boldsymbol{x}}_{t-1}|\boldsymbol{x}_{t+1})$ contains two consecutive sampling steps: $q_{\boldsymbol{\theta}}(\hat{\boldsymbol{x}}_t|\boldsymbol{x}_{t+1})$ and $q_{\boldsymbol{\theta}}(\hat{\boldsymbol{x}}_{t-1}|\hat{\boldsymbol{x}}_t)$, we can solve out $q_{\boldsymbol{\theta}}(\hat{\boldsymbol{x}}_{t-1}|\boldsymbol{x}_{t+1})$ by iterative plugging-in. According to $q_{\boldsymbol{\theta}}(\hat{\boldsymbol{x}}_t|\boldsymbol{x}_{t+1}) = \mathcal{N}(\hat{\boldsymbol{x}}_t; \mu_{\boldsymbol{\theta}}(\boldsymbol{x}_{t+1}, t+1), \tilde{\beta}_{t+1}\boldsymbol{I})$ and Eq. 11, we know that $q_{\boldsymbol{\theta}}(\hat{\boldsymbol{x}}_{t-1}|\hat{\boldsymbol{x}}_t) = \mathcal{N}(\hat{\boldsymbol{x}}_{t-1}; \mu_{\boldsymbol{\theta}}(\hat{\boldsymbol{x}}_t, t), \tilde{\beta}_t\boldsymbol{I})$ and a sample from $q_{\boldsymbol{\theta}}(\hat{\boldsymbol{x}}_{t-1}|\hat{\boldsymbol{x}}_t)$ is:

$$\hat{\boldsymbol{x}}_{t-1} = \frac{\sqrt{\bar{\alpha}_{t-1}}\beta_t}{1 - \bar{\alpha}_t}\boldsymbol{x}_{\boldsymbol{\theta}}^t + \frac{\sqrt{\alpha_t}(1 - \bar{\alpha}_{t-1})}{1 - \bar{\alpha}_t}\hat{\boldsymbol{x}}_t + \sqrt{\tilde{\beta}_t}\boldsymbol{\epsilon}_1. \tag{16}$$

From Table 1, we know that $q_{\boldsymbol{\theta}}(\hat{\boldsymbol{x}}_t|\boldsymbol{x}_{t+1}) = \mathcal{N}(\hat{\boldsymbol{x}}_t; \sqrt{\bar{\alpha}_t}\boldsymbol{x}_0, (1 - \bar{\alpha}_t + (\frac{\sqrt{\bar{\alpha}_t}\beta_{t+1}}{1 - \bar{\alpha}_{t+1}}e_{t+1})^2)\boldsymbol{I})$, so plug in $\hat{\boldsymbol{x}}_t = \sqrt{\bar{\alpha}_t}\boldsymbol{x}_0 + \sqrt{1 - \bar{\alpha}_t + (\frac{\sqrt{\bar{\alpha}_t}\beta_{t+1}}{1 - \bar{\alpha}_{t+1}}e_{t+1})^2}\boldsymbol{\epsilon}_3$ into Eq. 16, we know a sample from $q_{\boldsymbol{\theta}}(\hat{\boldsymbol{x}}_{t-1}|\boldsymbol{x}_{t+1})$ is:

$$\hat{\boldsymbol{x}}_{t-1} = \frac{\sqrt{\bar{\alpha}_{t-1}}\beta_t}{1 - \bar{\alpha}_t}\boldsymbol{x}_{\boldsymbol{\theta}}^t + \frac{\sqrt{\alpha_t}(1 - \bar{\alpha}_{t-1})}{1 - \bar{\alpha}_t}(\sqrt{\bar{\alpha}_t}\boldsymbol{x}_0 + \sqrt{1 - \bar{\alpha}_t + (\frac{\sqrt{\bar{\alpha}_t}\beta_{t+1}}{1 - \bar{\alpha}_{t+1}}e_{t+1})^2}\boldsymbol{\epsilon}_3) + \sqrt{\tilde{\beta}_t}\boldsymbol{\epsilon}_1. \tag{17}$$

By denoting $(\frac{\sqrt{\bar{\alpha}_t}\beta_{t+1}}{1 - \bar{\alpha}_{t+1}}e_{t+1})^2$ as $f(t)$ and plugging in $\boldsymbol{x}_{\boldsymbol{\theta}}^t = \boldsymbol{x}_0 + e_t\boldsymbol{\epsilon}_0$ (Eq. 7), we have:

$$\hat{\boldsymbol{x}}_{t-1} = \frac{\sqrt{\bar{\alpha}_{t-1}}\beta_t}{1 - \bar{\alpha}_t}(\boldsymbol{x}_0 + e_t\boldsymbol{\epsilon}_0) + \frac{\sqrt{\alpha_t}(1 - \bar{\alpha}_{t-1})}{1 - \bar{\alpha}_t}(\sqrt{\bar{\alpha}_t}\boldsymbol{x}_0 + \sqrt{1 - \bar{\alpha}_t + f(t)}\boldsymbol{\epsilon}_3) + \sqrt{\tilde{\beta}_t}\boldsymbol{\epsilon}_1 \tag{18}$$

$$\approx \frac{\sqrt{\bar{\alpha}_{t-1}}\beta_t}{1 - \bar{\alpha}_t}(\boldsymbol{x}_0 + e_t\boldsymbol{\epsilon}_0) + \frac{\sqrt{\alpha_t}(1 - \bar{\alpha}_{t-1})}{1 - \bar{\alpha}_t}(\sqrt{\bar{\alpha}_t}\boldsymbol{x}_0 + \sqrt{1 - \bar{\alpha}_t}\boldsymbol{\epsilon}_3 + \frac{1}{2\sqrt{1 - \bar{\alpha}_t}}f(t)\boldsymbol{\epsilon}_3) + \sqrt{\tilde{\beta}_t}\boldsymbol{\epsilon}_1 \tag{19}$$

$$\approx \frac{\sqrt{\bar{\alpha}_{t-1}}\beta_t}{1 - \bar{\alpha}_t}\boldsymbol{x}_0 + \frac{\sqrt{\bar{\alpha}_{t-1}}\beta_t}{1 - \bar{\alpha}_t}e_t\boldsymbol{\epsilon}_0 + \frac{\sqrt{\alpha_t}(1 - \bar{\alpha}_{t-1})}{1 - \bar{\alpha}_t}\sqrt{\bar{\alpha}_t}\boldsymbol{x}_0$$
$$+ \frac{\sqrt{\alpha_t}(1 - \bar{\alpha}_{t-1})}{1 - \bar{\alpha}_t}(\sqrt{1 - \bar{\alpha}_t}\boldsymbol{\epsilon}_3 + \frac{1}{2\sqrt{1 - \bar{\alpha}_t}}f(t)\boldsymbol{\epsilon}_3) + \sqrt{\tilde{\beta}_t}\boldsymbol{\epsilon}_1 \tag{20}$$

$$\approx \sqrt{\bar{\alpha}_{t-1}}\boldsymbol{x}_0 + \frac{\sqrt{\bar{\alpha}_{t-1}}\beta_t}{1 - \bar{\alpha}_t}e_t\boldsymbol{\epsilon}_0 + \frac{\sqrt{\alpha_t}(1 - \bar{\alpha}_{t-1})}{1 - \bar{\alpha}_t}(\sqrt{1 - \bar{\alpha}_t} + \frac{1}{2\sqrt{1 - \bar{\alpha}_t}}f(t))\boldsymbol{\epsilon}_3 + \sqrt{\tilde{\beta}_t}\boldsymbol{\epsilon}_1 \tag{21}$$

Taylor's theorem is used from Eq. 18 to Eq. 19. The process from Eq. 20 to Eq. 21 is similar to the simplification from Eq. 12 to Eq. 13. From Eq. 21, we know that the mean of $q_{\boldsymbol{\theta}}(\hat{\boldsymbol{x}}_{t-1}|\boldsymbol{x}_{t+1})$ is $\sqrt{\bar{\alpha}_{t-1}}\boldsymbol{x}_0$. We now focus on the variance:

$$Var(\hat{\boldsymbol{x}}_{t-1}) = (\frac{\sqrt{\bar{\alpha}_{t-1}}\beta_t}{1 - \bar{\alpha}_t}e_t)^2 + (\frac{\sqrt{\alpha_t}(1 - \bar{\alpha}_{t-1})}{1 - \bar{\alpha}_t}\sqrt{1 - \bar{\alpha}_t})^2 + \tilde{\beta}_t$$
$$+ (\frac{\sqrt{\alpha_t}(1 - \bar{\alpha}_{t-1})}{1 - \bar{\alpha}_t}\frac{1}{2\sqrt{1 - \bar{\alpha}_t}}f(t))^2 \tag{22}$$

$$= 1 - \bar{\alpha}_{t-1} + (\frac{\sqrt{\bar{\alpha}_{t-1}}\beta_t}{1 - \bar{\alpha}_t}e_t)^2 + \frac{\alpha_t(1 - \bar{\alpha}_{t-1})^2}{4(1 - \bar{\alpha}_t)^3}f(t)^2 \tag{23}$$

The above derivation is similar to the progress from Eq. 14 to Eq. 15. Now we write the mean and variance of $q_{\boldsymbol{\theta}}(\hat{\boldsymbol{x}}_{t-1}|\boldsymbol{x}_{t+1})$ in Table 7. In the same spirit of iterative plugging-in, we could derive $(\hat{\boldsymbol{x}}_t|\boldsymbol{x}_T)$ which has the mean $\sqrt{\bar{\alpha}_t}\boldsymbol{x}_0$ and variance larger than $(1 - \bar{\alpha}_t)\boldsymbol{I}$.

## A.3 DERIVATION OF $q_{\boldsymbol{\theta}}(\hat{\boldsymbol{x}}_t|\boldsymbol{x}_{t+1})$ FOR DDIM

We first review the derivation of the reverse diffusion $p_{\boldsymbol{\theta}}(\boldsymbol{x}_{t-1}|\boldsymbol{x}_t)$ for DDIM. To keep the symbols consistent in this paper, we continue to use the notations of DDPM in the derivation of DDIM.

Table 7: The distribution $q(\boldsymbol{x}_{t-1}|\boldsymbol{x}_0)$ during training and $q_{\boldsymbol{\theta}}(\hat{\boldsymbol{x}}_{t-1}|\boldsymbol{x}_{t+1})$ during DDPM sampling.

|  | Mean | Variance |
|---|---|---|
| $q(\boldsymbol{x}_{t-1}|\boldsymbol{x}_0)$ | $\sqrt{\bar{\alpha}_{t-1}}\boldsymbol{x}_0$ | $(1-\bar{\alpha}_{t-1})\boldsymbol{I}$ |
| $q_{\boldsymbol{\theta}}(\hat{\boldsymbol{x}}_{t-1}|\boldsymbol{x}_{t+1})$ | $\sqrt{\bar{\alpha}_{t-1}}\boldsymbol{x}_0$ | $(1-\bar{\alpha}_{t-1}+(\frac{\sqrt{\bar{\alpha}_{t-1}}\beta_t}{1-\bar{\alpha}_t}e_t)^2 + \frac{\alpha_t(1-\bar{\alpha}_{t-1})^2}{4(1-\bar{\alpha}_t)^3}f(t)^2)\boldsymbol{I}$ |

Recall that DDIM and DDPM have the same loss function because they share the same marginal distribution $q(\boldsymbol{x}_t|\boldsymbol{x}_0) = \mathcal{N}(\boldsymbol{x}_t; \sqrt{\bar{\alpha}_t}\boldsymbol{x}_0, (1-\bar{\alpha}_t)\boldsymbol{I})$. But the posterior $q(\boldsymbol{x}_{t-1}|\boldsymbol{x}_t, \boldsymbol{x}_0)$ of DDIM is obtained under Non-Markovian diffusion process and is given by Song et al. (2021a):

$$q(\boldsymbol{x}_{t-1}|\boldsymbol{x}_t, \boldsymbol{x}_0) = \mathcal{N}(\sqrt{\bar{\alpha}_{t-1}}\boldsymbol{x}_0 + \sqrt{1-\bar{\alpha}_{t-1}-\sigma_t^2} \cdot \frac{\boldsymbol{x}_t - \sqrt{\bar{\alpha}_t}\boldsymbol{x}_0}{\sqrt{1-\bar{\alpha}_t}}, \sigma_t^2\boldsymbol{I}). \tag{24}$$

Similar to DDPM, the reverse distribution of DDIM is parameterized as $p_{\boldsymbol{\theta}}(\boldsymbol{x}_{t-1}|\boldsymbol{x}_t) = q(\boldsymbol{x}_{t-1}|\boldsymbol{x}_t, \boldsymbol{x}_{\boldsymbol{\theta}}^t)$, where $\boldsymbol{x}_{\boldsymbol{\theta}}^t$ means the predicted $\boldsymbol{x}_0$ given $\boldsymbol{x}_t$. Based on Eq. 24, the reverse diffusion $q(\boldsymbol{x}_{t-1}|\boldsymbol{x}_t, \boldsymbol{x}_{\boldsymbol{\theta}}^t)$ is:

$$q(\boldsymbol{x}_{t-1}|\boldsymbol{x}_t, \boldsymbol{x}_{\boldsymbol{\theta}}^t) = \mathcal{N}(\sqrt{\bar{\alpha}_{t-1}}\boldsymbol{x}_{\boldsymbol{\theta}}^t + \sqrt{1-\bar{\alpha}_{t-1}-\sigma_t^2} \cdot \frac{\boldsymbol{x}_t - \sqrt{\bar{\alpha}_t}\boldsymbol{x}_{\boldsymbol{\theta}}^t}{\sqrt{1-\bar{\alpha}_t}}, \sigma_t^2\boldsymbol{I}). \tag{25}$$

Again, we point out that $q(\boldsymbol{x}_{t-1}|\boldsymbol{x}_t, \boldsymbol{x}_{\boldsymbol{\theta}}^t) = q(\boldsymbol{x}_{t-1}|\boldsymbol{x}_t, \boldsymbol{x}_0)$ holds only if $\boldsymbol{x}_{\boldsymbol{\theta}}^t = \boldsymbol{x}_0$, this requires the network to make no prediction error about $\boldsymbol{x}_0$. Theoretically, we need to consider the uncertainty of the prediction $\boldsymbol{x}_{\boldsymbol{\theta}}^t$ and model it as a probabilistic distribution $p_{\boldsymbol{\theta}}(\boldsymbol{x}_0|\boldsymbol{x}_t)$. Following Analytical-DPM (Bao et al., 2022), we approximate it by a Gaussian distribution $p_{\boldsymbol{\theta}}(\boldsymbol{x}_0|\boldsymbol{x}_t) = \mathcal{N}(\boldsymbol{x}_{\boldsymbol{\theta}}^t; \boldsymbol{x}_0, e_t^2\boldsymbol{I})$, namely $\boldsymbol{x}_{\boldsymbol{\theta}}^t = \boldsymbol{x}_0 + e_t\boldsymbol{\epsilon}_0$. Thus, the practical reverse diffusion $q(\boldsymbol{x}_{t-1}|\boldsymbol{x}_t, \boldsymbol{x}_{\boldsymbol{\theta}}^t)$ is

$$q(\boldsymbol{x}_{t-1}|\boldsymbol{x}_t, \boldsymbol{x}_{\boldsymbol{\theta}}^t) = \mathcal{N}(\sqrt{\bar{\alpha}_{t-1}}(\boldsymbol{x}_0 + e_t\boldsymbol{\epsilon}_0) + \sqrt{1-\bar{\alpha}_{t-1}-\sigma_t^2} \cdot \frac{\boldsymbol{x}_t - \sqrt{\bar{\alpha}_t}(\boldsymbol{x}_0 + e_t\boldsymbol{\epsilon}_0)}{\sqrt{1-\bar{\alpha}_t}}, \sigma_t^2\boldsymbol{I}). \tag{26}$$

Note that $\sigma_t = 0$ for DDIM sampler, so a sample $\boldsymbol{x}_{t-1}$ from $q(\boldsymbol{x}_{t-1}|\boldsymbol{x}_t, \boldsymbol{x}_{\boldsymbol{\theta}}^t)$ is:

$$\boldsymbol{x}_{t-1} = \sqrt{\bar{\alpha}_{t-1}}(\boldsymbol{x}_0 + e_t\boldsymbol{\epsilon}_0) + \sqrt{1-\bar{\alpha}_{t-1}-\sigma_t^2} \cdot \frac{\boldsymbol{x}_t - \sqrt{\bar{\alpha}_t}(\boldsymbol{x}_0 + e_t\boldsymbol{\epsilon}_0)}{\sqrt{1-\bar{\alpha}_t}} + \sigma_t\boldsymbol{\epsilon}_4$$

$$= \sqrt{\bar{\alpha}_{t-1}}\boldsymbol{x}_0 + \sqrt{\bar{\alpha}_{t-1}}e_t\boldsymbol{\epsilon}_0 + \sqrt{1-\bar{\alpha}_{t-1}} \cdot \frac{\boldsymbol{x}_t - \sqrt{\bar{\alpha}_t}\boldsymbol{x}_0}{\sqrt{1-\bar{\alpha}_t}} - \sqrt{1-\bar{\alpha}_{t-1}} \cdot \frac{\sqrt{\bar{\alpha}_t}e_t\boldsymbol{\epsilon}_0}{\sqrt{1-\bar{\alpha}_t}} \tag{27}$$

$$= \sqrt{\bar{\alpha}_{t-1}}\boldsymbol{x}_0 + \sqrt{\bar{\alpha}_{t-1}}e_t\boldsymbol{\epsilon}_0 + \sqrt{1-\bar{\alpha}_{t-1}}\boldsymbol{\epsilon}_5 - \sqrt{1-\bar{\alpha}_{t-1}} \cdot \frac{\sqrt{\bar{\alpha}_t}e_t\boldsymbol{\epsilon}_0}{\sqrt{1-\bar{\alpha}_t}} \tag{28}$$

$$= \sqrt{\bar{\alpha}_{t-1}}\boldsymbol{x}_0 + \sqrt{1-\bar{\alpha}_{t-1}}\boldsymbol{\epsilon}_5 + (\sqrt{\bar{\alpha}_{t-1}}e_t - \sqrt{1-\bar{\alpha}_{t-1}} \cdot \frac{\sqrt{\bar{\alpha}_t}e_t}{\sqrt{1-\bar{\alpha}_t}})\boldsymbol{\epsilon}_0 \tag{29}$$

From Eq. 27 to Eq. 28, we plug in $\boldsymbol{x}_t = \sqrt{\bar{\alpha}_t}\boldsymbol{x}_0 + \sqrt{1-\bar{\alpha}_t}\boldsymbol{\epsilon}_5$ where $\boldsymbol{\epsilon}_5 \sim \mathcal{N}(\boldsymbol{0}, \boldsymbol{I})$. We now compute the sampling distribution $q(\hat{\boldsymbol{x}}_t|\boldsymbol{x}_{t+1}, \boldsymbol{x}_{\boldsymbol{\theta}}^{t+1})$ which is the same distribution as $q(\boldsymbol{x}_{t-1}|\boldsymbol{x}_t, \boldsymbol{x}_{\boldsymbol{\theta}}^t)$ by replacing the index $t$ with $t+1$ and using $\hat{\boldsymbol{x}}_t$ to highlight it is a generated sample. According to Eq. 29, a sample $\hat{\boldsymbol{x}}_t$ from $q(\hat{\boldsymbol{x}}_t|\boldsymbol{x}_{t+1}, \boldsymbol{x}_{\boldsymbol{\theta}}^{t+1})$ is:

$$\hat{\boldsymbol{x}}_t = \sqrt{\bar{\alpha}_t}\boldsymbol{x}_0 + \sqrt{1-\bar{\alpha}_t}\boldsymbol{\epsilon}_5 + (\sqrt{\bar{\alpha}_t}e_{t+1} - \sqrt{1-\bar{\alpha}_t} \cdot \frac{\sqrt{\bar{\alpha}_{t+1}}e_{t+1}}{\sqrt{1-\bar{\alpha}_{t+1}}})\boldsymbol{\epsilon}_0 \tag{30}$$

From Eq. 30, we know the mean of $q(\hat{\boldsymbol{x}}_t|\boldsymbol{x}_{t+1}, \boldsymbol{x}_{\boldsymbol{\theta}}^{t+1})$ is $\sqrt{\bar{\alpha}_t}\boldsymbol{x}_0$. We now calculate the variance by looking at $\sqrt{1-\bar{\alpha}_t}\boldsymbol{\epsilon}_5 + (\sqrt{\bar{\alpha}_t}e_{t+1} - \sqrt{1-\bar{\alpha}_t} \cdot \frac{\sqrt{\bar{\alpha}_{t+1}}e_{t+1}}{\sqrt{1-\bar{\alpha}_{t+1}}})\boldsymbol{\epsilon}_0$:

$$
\begin{aligned}
Var(\hat{\boldsymbol{x}}_t) &= (\sqrt{1-\bar{\alpha}_t})^2 + (\sqrt{\bar{\alpha}_t}e_{t+1} - \sqrt{1-\bar{\alpha}_t} \cdot \frac{\sqrt{\bar{\alpha}_{t+1}}e_{t+1}}{\sqrt{1-\bar{\alpha}_{t+1}}})^2 \\
&= 1 - \bar{\alpha}_t + (\sqrt{\bar{\alpha}_t} - \sqrt{1-\bar{\alpha}_t} \cdot \frac{\sqrt{\bar{\alpha}_{t+1}}}{\sqrt{1-\bar{\alpha}_{t+1}}})^2 e_{t+1}^2 \\
&= 1 - \bar{\alpha}_t + (\sqrt{\bar{\alpha}_t} - \frac{\sqrt{1-\bar{\alpha}_t}\sqrt{\bar{\alpha}_t}\sqrt{\alpha_{t+1}}}{\sqrt{1-\bar{\alpha}_{t+1}}})^2 e_{t+1}^2 \\
&= 1 - \bar{\alpha}_t + (\sqrt{\bar{\alpha}_t}(1 - \frac{\sqrt{1-\bar{\alpha}_t}\sqrt{\alpha_{t+1}}}{\sqrt{1-\bar{\alpha}_{t+1}}}))^2 e_{t+1}^2 \\
&= 1 - \bar{\alpha}_t + \bar{\alpha}_t(1 - \frac{\sqrt{\alpha_{t+1}-\bar{\alpha}_{t+1}}}{\sqrt{1-\bar{\alpha}_{t+1}}})^2 e_{t+1}^2 \\
&= 1 - \bar{\alpha}_t + (1 - \sqrt{\frac{\alpha_{t+1}-\bar{\alpha}_{t+1}}{1-\bar{\alpha}_{t+1}}})^2 \bar{\alpha}_t e_{t+1}^2
\end{aligned}
\tag{31}
$$

As a result, we can write the mean and variance of the sampling distribution $q(\hat{\boldsymbol{x}}_t|\boldsymbol{x}_{t+1}, \boldsymbol{x}_{\boldsymbol{\theta}}^{t+1})$, i.e. $q_{\boldsymbol{\theta}}(\hat{\boldsymbol{x}}_t|\boldsymbol{x}_{t+1})$, and compare it with the training distribution $q(\boldsymbol{x}_t|\boldsymbol{x}_0)$ in Table 8.

Table 8: The mean and variance of $q(\boldsymbol{x}_t|\boldsymbol{x}_0)$ during training and $q_{\boldsymbol{\theta}}(\hat{\boldsymbol{x}}_t|\boldsymbol{x}_{t+1})$ during DDIM sampling.

| | Mean | Variance |
|---|---|---|
| $q(\boldsymbol{x}_t|\boldsymbol{x}_0)$ | $\sqrt{\bar{\alpha}_t}\boldsymbol{x}_0$ | $(1-\bar{\alpha}_t)\boldsymbol{I}$ |
| $q_{\boldsymbol{\theta}}(\hat{\boldsymbol{x}}_t|\boldsymbol{x}_{t+1})$ | $\sqrt{\bar{\alpha}_t}\boldsymbol{x}_0$ | $(1-\bar{\alpha}_t + (1 - \sqrt{\frac{\alpha_{t+1}-\bar{\alpha}_{t+1}}{1-\bar{\alpha}_{t+1}}})^2 \bar{\alpha}_t e_{t+1}^2)\boldsymbol{I}$ |

Since $\alpha_{t+1} < 1$, $\sqrt{\frac{\alpha_{t+1}-\bar{\alpha}_{t+1}}{1-\bar{\alpha}_{t+1}}} < 1$ and $(1 - \sqrt{\frac{\alpha_{t+1}-\bar{\alpha}_{t+1}}{1-\bar{\alpha}_{t+1}}}) > 0$ hold for any $t$ in Eq. 31. Similar to DDPM sampler, the variance of $q(\hat{\boldsymbol{x}}_t|\boldsymbol{x}_{t+1}, \boldsymbol{x}_{\boldsymbol{\theta}}^{t+1})$ is always larger than that of $q(\boldsymbol{x}_t|\boldsymbol{x}_0)$ by the magnitude $(1 - \sqrt{\frac{\alpha_{t+1}-\bar{\alpha}_{t+1}}{1-\bar{\alpha}_{t+1}}})^2 \bar{\alpha}_t e_{t+1}^2$, indicating the exposure bias issue in DDIM sampler.

## A.4 Practical Variance Error of $q_{\boldsymbol{\theta}}(\hat{\boldsymbol{x}}_t|\boldsymbol{x}_{t+1})$ and $q_{\boldsymbol{\theta}}(\hat{\boldsymbol{x}}_t|\boldsymbol{x}_T)$

We measure the single-step variance error of $q_{\boldsymbol{\theta}}(\hat{\boldsymbol{x}}_t|\boldsymbol{x}_{t+1})$ and multi-step variance error of $q_{\boldsymbol{\theta}}(\hat{\boldsymbol{x}}_t|\boldsymbol{x}_T)$ using Algorithm 1 and Algorithm 2, respectively. Note that, the multi-step variance error measurement is similar to the exposure bias $\delta_t$ evaluation and we denote the single-step variance error as $\Delta_t$ and represent the multi-step variance error as $\Delta_t'$. The experiments are implemented on CIFAR-10 (Krizhevsky et al., 2009) dataset and ADM model (Dhariwal & Nichol, 2021). The key difference between $\Delta_t$ and $\Delta_t'$ measurement is that the former can get access to the ground truth input $\boldsymbol{x}_t$ at each sampling step $t$, while the latter is only exposed to the predicted $\hat{\boldsymbol{x}}_t$ in the iterative sampling process.

**Algorithm 1** Variance error under single-step sampling

1: Initialize $\Delta_t = 0$, $n_t = list()$ ($\forall t \in \{1, ..., T-1\}$)
2: **for** $t := T, ..., 1$ **do**
3:     **repeat**
4:       $\boldsymbol{x}_0 \sim q(\boldsymbol{x}_0), \boldsymbol{\epsilon} \sim \mathcal{N}(\boldsymbol{0}, \boldsymbol{I})$
5:       $\boldsymbol{x}_t = \sqrt{\bar{\alpha}_t}\boldsymbol{x}_0 + \sqrt{1-\bar{\alpha}_t}\boldsymbol{\epsilon}$
6:       $\hat{\boldsymbol{x}}_{t-1} = \frac{1}{\sqrt{\alpha_t}}(\boldsymbol{x}_t - \frac{\beta_t}{\sqrt{1-\bar{\alpha}_t}}\boldsymbol{\epsilon_\theta}(\boldsymbol{x}_t, t)) +$
        $\sqrt{\tilde{\beta}_t}\boldsymbol{z}$   $(\boldsymbol{z} \sim \mathcal{N}(\boldsymbol{0}, \boldsymbol{I}))$
7:       $n_{t-1}.append(\hat{\boldsymbol{x}}_{t-1} - \sqrt{\bar{\alpha}_{t-1}}\boldsymbol{x}_0)$
8:     **until** 50k iterations
9: **end for**
10: **for** $t := T, ..., 1$ **do**
11:     $\hat{\beta}_t = numpy.var(n_t)$
12:     $\Delta_t = \hat{\beta}_t - \bar{\beta}_t$
13: **end for**

**Algorithm 2** Variance error under multi-step sampling

1: Initialize $\delta_t = 0$, $n_t = list()$ ($\forall t \in \{1, ..., T-1\}$)
2: **repeat**
3:     $\boldsymbol{x}_0 \sim q(\boldsymbol{x}_0), \boldsymbol{\epsilon} \sim \mathcal{N}(\boldsymbol{0}, \boldsymbol{I})$
4:     $\boldsymbol{x}_T = \sqrt{\bar{\alpha}_T}\boldsymbol{x}_0 + \sqrt{1-\bar{\alpha}_T}\boldsymbol{\epsilon}$
5:     **for** $t := T, ..., 1$ **do**
6:       if $t == T$ then $\hat{\boldsymbol{x}}_t = \boldsymbol{x}_T$
7:       $\hat{\boldsymbol{x}}_{t-1} = \frac{1}{\sqrt{\alpha_t}}(\hat{\boldsymbol{x}}_t - \frac{\beta_t}{\sqrt{1-\bar{\alpha}_t}}\boldsymbol{\epsilon_\theta}(\hat{\boldsymbol{x}}_t, t)) +$
        $\sqrt{\tilde{\beta}_t}\boldsymbol{z}$   $(\boldsymbol{z} \sim \mathcal{N}(\boldsymbol{0}, \boldsymbol{I}))$
8:       $n_{t-1}.append(\hat{\boldsymbol{x}}_{t-1} - \sqrt{\bar{\alpha}_{t-1}}\boldsymbol{x}_0)$
9:     **end for**
10: **until** 50k iterations
11: **for** $t := T, ..., 1$ **do**
12:     $\hat{\beta}_t = numpy.var(n_t)$
13:     $\Delta'_t = \hat{\beta}_t - \bar{\beta}_t$
14: **end for**

### A.5 METRIC FOR EXPOSURE BIAS

The key step of Algorithm 3 is that we subtract the mean $\sqrt{\bar{\alpha}_{t-1}}\boldsymbol{x}_0$ and the remaining term $\hat{\boldsymbol{x}}_{t-1} - \sqrt{\bar{\alpha}_{t-1}}\boldsymbol{x}_0$ corresponds to the stochastic term of $q(\hat{\boldsymbol{x}}_{t-1}|\boldsymbol{x}_t, \boldsymbol{x}_{\boldsymbol{\theta}}^t)$. In our experiments, we use $N = 50,000$ samples to compute the variance $\hat{\bar{\beta}}_t$.

**Algorithm 3** Measurement of Exposure Bias $\delta_t$

1: Initialize $\delta_t = 0$, $n_t = list()$ ($\forall t \in \{1, ..., T-1\}$)
2: **repeat**
3:     $\boldsymbol{x}_0 \sim q(\boldsymbol{x}_0), \boldsymbol{\epsilon} \sim \mathcal{N}(\boldsymbol{0}, \boldsymbol{I})$
4:     compute $\boldsymbol{x}_T$ using Eq. 1
5:     **for** $t := T, ..., 1$ **do**
6:       if $t == T$ then $\hat{\boldsymbol{x}}_t = \boldsymbol{x}_T$
7:       $\hat{\boldsymbol{x}}_{t-1} = \frac{1}{\sqrt{\alpha_t}}(\hat{\boldsymbol{x}}_t - \frac{\beta_t}{\sqrt{1-\bar{\alpha}_t}}\boldsymbol{\epsilon_\theta}(\hat{\boldsymbol{x}}_t, t)) + \sqrt{\tilde{\beta}_t}\boldsymbol{z}$   $(\boldsymbol{z} \sim \mathcal{N}(\boldsymbol{0}, \boldsymbol{I}))$
8:       $n_{t-1}.append(\hat{\boldsymbol{x}}_{t-1} - \sqrt{\bar{\alpha}_{t-1}}\boldsymbol{x}_0)$
9:     **end for**
10: **until** $N$ iterations
11: **for** $t := T, ..., 1$ **do**
12:     $\hat{\beta}_t = numpy.var(n_t)$
13:     $\delta_t = (\sqrt{\hat{\beta}_t} - \sqrt{\bar{\beta}_t})^2$
14: **end for**

### A.6 CORRELATION BETWEEN EXPOSURE BIAS METRIC AND FID

We define the exposure bias at timestep $t$ as $\delta_t = (\sqrt{\beta'_t} - \sqrt{\bar{\beta}_t})^2$, where $\bar{\beta}_t = 1 - \bar{\alpha}_t$ denotes the variance of $q(\boldsymbol{x}_t|\boldsymbol{x}_0)$ during training and $\beta'_t$ presents the variance of $q_{\boldsymbol{\theta}}(\hat{\boldsymbol{x}}_t|\boldsymbol{x}_T)$ in the regular sampling process. Although $\delta_t$ measures the discrepancy between network inputs and FID to evaluate the difference between training data and network outputs, we empirically find a strong correlation between $\delta_t$ and FID, which could arise from the benefit of defining $\delta_t$ from the Fréchet distance Dowson & Landau

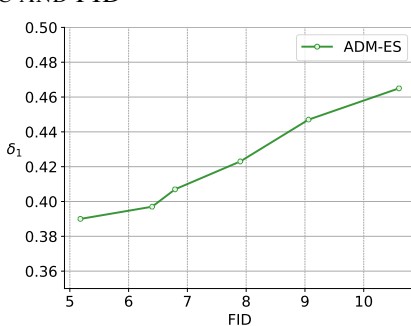

Figure 8: Correlation between FID - $\delta_1$.

(1982) perspective. In Fig. 8, we present the FID-$\delta_1$ relationships on CIFAR-10 and use 20-step sampling, wherein $\delta_1$ represents the exposure bias in the last sampling step $t = 1$. Additionally, $\delta_t$ has the advantage of indicating the network input quality at any intermediate timestep $t$. Taking Fig. 4 as an example, we can see that the input quality decreases dramatically near the end of sampling ($t = 1$) as $\delta_t$ increases significantly.

## A.7 FID RESULTS ON PFGM++ BASELINE

In PFGM, Xu et al. (2022) claim that the strong norm-t correlation causes the sensitivity of prediction errors in diffusion frameworks, and PFGM is more resistant to prediction errors due to the greater range of training sample norms. Our experiments verify the robustness of Poisson Flow frameworks by observing that Epsilon Scaling enjoys a wider range of $\lambda(t)$ on PFGM++ baseline (Xu et al., 2023b) than on EDM baseline. The rationale is simple: an inappropriate $\lambda((t))$ introduces errors that require the network to be robust to counteract. However, we observe that the exposure bias issue still exists in Poisson Flow frameworks even though they are less sensitive to prediction errors than diffusion frameworks. Therefore, Epsilon Scaling is applicable in Poisson Flow models. Table 9 and Table 10 show the significant improvements made by Epsilon Scaling ($\lambda(t) = b$) on PFGM++.

Table 9: FID on CIFAR-10 using PFGM++ baseline (D=128, uncond).

| Model | $T'$ | | | |
|---|---|---|---|---|
| | 9 | 13 | 21 | 35 |
| PFGM++ | 37.82 | 7.55 | 2.34 | 1.92 |
| PFGM++ with ES | **17.91** | **4.51** | **2.31** | **1.91** |
| | **(b=1.008)** | **(b=1.016)** | **(b=0.970)** | **(b=1.00045)** |

Table 10: FID on CIFAR-10 using PFGM++ baseline (D=2048, uncond).

| Model | $T'$ | | | |
|---|---|---|---|---|
| | 9 | 13 | 21 | 35 |
| PFGM++ | 37.16 | 7.34 | 2.31 | 1.91 |
| PFGM++ with ES | **17.27** | **4.88** | **2.20** | **1.90** |
| | **(b=1.007)** | **(b=1.015)** | **(b=0.996)** | **(b=1.0007)** |

## A.8 FID RESULTS UNDER $T' > 100$

Although using sampling step $T' = 100$ often achieves the near-optimal FID in diffusion models, we still report the performance of our Epsilon Scaling in the large $T'$ regions. We show the FID results on Analytic-DDPM (Bao et al., 2022) baseline (Table 11) and ADM baseline (Table 12), where we apply $\lambda(t) = b$ for Epsilon Scaling.

Table 11: FID on CIFAR-10 using Analytic-DDPM baseline (linear noise schedule).

| | $T'$ | | | | | |
|---|---|---|---|---|---|---|
| | 20 | 50 | 100 | 200 | 400 | 1000 |
| Analytic-DDPM | 14.61 | 7.25 | 5.40 | 4.01 | 3.62 | 4.03 |
| Analytic-DDPM with ES | **11.02** | **5.03** | **4.09** | **3.39** | **3.14** | **3.42** |

Table 12: FID on CIFAR-10 using ADM baseline.

| | $T'$ | | |
|---|---|---|---|
| | 200 | 300 | 1000 |
| ADM | 3.04 | 2.95 | 3.01 |
| ADM-ES | **2.15** | **2.14** | **2.21** |

## A.9 RECALL AND PRECISION RESULTS

Our method Epsilon Scaling does not affect the recall and precision of the base model. We present the complete recall and precision (Kynkäänniemi et al., 2019) results in Table 13 using the code provided by ADM (Dhariwal & Nichol, 2021). ADM-ES achieve higher recalls and slightly lower previsions across the five datasets. But the overall differences are minor.

Table 13: Recall and precision of ADM and ADM-ES using 100-step sampling.

| Model | CIFAR-10 32×32 | | LSUN tower 64×64 | | FFHQ 128×128 | | ImageNet 64×64 | | ImageNet 128×128 | |
|---|---|---|---|---|---|---|---|---|---|---|
| | recall | precision | recall | precision | recall | precision | recall | precision | recall | precision |
| ADM | 0.591 | **0.691** | 0.605 | **0.645** | 0.497 | **0.696** | 0.621 | **0.738** | 0.586 | 0.771 |
| ADM-ES | **0.613** | 0.684 | **0.606** | 0.641 | **0.545** | 0.683 | **0.632** | 0.726 | **0.592** | 0.771 |

## A.10 EPSILON SCALING PARAMETERS: $k, b$

We present the parameters $k, b$ of Epsilon Scaling we used in all of our experiments in Table 14, Table 15 and Table 16 for reproducibility. Apart from that, we provide suggestions on finding the optimal parameters even though they are dependent on the dataset and how well the base model is trained. Our suggestions are:

- Search for the optimal uniform schedule $\lambda(t) = b$ in a coarse-to-fine manner: use stride 0.001, 0.0005, 0.0001 progressively.
- In general, the optimal $b$ will decrease as the sampling step $T'$ increases.
- After getting the optimal uniform schedule $\lambda(t)^* = b^*$, we search for $k$ of the linear schedule $\lambda(t) = kt + b$ by keeping $\Sigma\lambda(t) = \Sigma\lambda(t)^*$, thereby, $b$ in the linear schedule is calculated rather than being searched.
- Instead of generating 50k samples, using 10k samples to compute FID for searching $\lambda(t)$.

Table 14: Epsilon Scaling schedule $\lambda(t) = kt + b$ we used on ADM baseline. We keep the FID results in the table for comparisons and remark $k, b$ underneath FIDs

| $T'$ | Model | Unconditional | | | Conditional | |
|---|---|---|---|---|---|---|
| | | CIFAR-10 32×32 | LSUN tower 64×64 | FFHQ 128×128 | ImageNet 64×64 | ImageNet 128×128 |
| 100 | ADM | 3.37 | 3.59 | 14.52 | 2.71 | 3.55 |
| | ADM-ES | 2.17 **(b=1.017)** | 2.91 **(b=1.006)** | 6.77 **(b=1.005)** | 2.39 **(b=1.006)** | 3.37 **(b=1.004)** |
| 50 | ADM | 4.43 | 7.28 | 26.15 | 3,75 | 5.15 |
| | ADM-ES | 2.49 **(b=1.017)** | 3.68 **(b=1.007)** | 9.50 **(b=1.007)** | 3.07 **(b=1.006)** | 4.33 **(b=1.004)** |
| 20 | ADM | 10.36 | 23.92 | 59.35 | 10.96 | 12.48 |
| | ADM-ES | 5.15 **(b=1.017)** | 8.22 **(b=1.011)** | 26.14 **(b=1.008)** | 7.52 **(b=1.006)** | 9.95 **(b=1.005)** |
| | ADM-ES* | 4.31 **(k=0.0025, b=1.0)** | 7.60 **(k=0.0008, b=1.0034)** | 24.83 **(k=0.0004, b=1.0042)** | 7.37 **(k=0.0002, b=1.0041)** | 9.86 **(k=0.00022, b=1.00291)** |

## A.11 INSENSITIVITY OF $\lambda(t)$

We empirically show that the FID gain can be achieved within a wide range of $\lambda(t)$. Table 17 and Table 18 demonstrate the insensitivity of our hyperparameters, which ease the search of $\lambda(t)$.

Table 15: Epsilon Scaling schedule $\lambda(t) = kt + b, (k = 0)$ we used on DDIM/DDPM and LDM baseline. We keep the FID results in the table for comparisons and remark $b$ underneath FIDs

| $T'$ | Model | CIFAR-10 32×32 | | CelebA 64×64 | |
|---|---|---|---|---|---|
| | | $\eta = 0$ | $\eta = 1$ | $\eta = 0$ | $\eta = 1$ |
| 100 | DDIM | 4.06 | 6.73 | 5.67 | 11.33 |
| | DDIM-ES | 3.38 **(b=1.0014)** | 4.01 **(b=1.03)** | 5.05 **(b=1.003)** | 4.45 **(b=1.04)** |
| 50 | DDIM | 4.82 | 10.29 | 6.88 | 15.09 |
| | DDIM-ES | 4.17 **(b=1.0030)** | 4.57 **(b=1.04)** | 6.20 **(b=1.004)** | 5.57 **(b=1.05)** |
| 20 | DDIM | 8.21 | 20.15 | 10.43 | 22.61 |
| | DDIM-ES | 6.54 **(b=1.0052)** | 7.78 **(b=1.05)** | 10.38 **(b=1.001)** | 11.83 **(b=1.06)** |

| $T'$ | Model | FFHQ 256×256 | CelebA-HQ 256×256 |
|---|---|---|---|
| 100 | LDM | 10.90 | 9.31 |
| | LDM-ES | 9.83 **(b=1.00015)** | 7.36 **(b=1.0009)** |
| 50 | LDM | 14.34 | 13.95 |
| | LDM-ES | 11.57 **(b=1.0016)** | 9.16 **(b=1.003)** |
| 20 | LDM | 33.13 | 29.62 |
| | LDM-ES | 20.91 **(b=1.007)** | 15.68 **(b=1.010)** |

Table 16: Epsilon Scaling schedule $\lambda(t) = kt + b, (k = 0)$ we used on EDM baseline. We keep the FID results in the table for comparisons and remark $b$ underneath FIDs

| $T'$ | Model | Unconditional | | Conditional | |
|---|---|---|---|---|---|
| | | Heun | Euler | Heun | Euler |
| 35 | EDM | 1.97 | 3.81 | 1.82 | 3.74 |
| | EDM-ES | 1.95 **b=1.0005** | 2.80 **b=1.0034** | 1.80 **b=1.0006** | 2.59 **b=1.0035** |
| 21 | EDM | 2.33 | 6.29 | 2.17 | 5.91 |
| | EDM-ES | 2.24 **b=0.9985** | 4.32 **b=1.0043** | 2.08 **b=0.9983** | 3.74 **b=1.0045** |
| 13 | EDM | 7.16 | 12.28 | 6.69 | 10.66 |
| | EDM-ES | 6.54 **b=1.0060** | 8.39 **b=1.0048** | 6.16 **b=1.0070** | 6.59 **b=1.0051** |

Table 17: FID of ADM-ES achieved on CIFAR-10 under different $\lambda(t)$ ($\lambda(t) = b$) and unconditional sampling, b=1 represents ADM.

| b | 1 (baseline) | 1.015 | 1.016 | 1.017 | 1.018 | 1.019 |
|---|---|---|---|---|---|---|
| $T' = 100$ | 3.37 | 2.20 | 2.18 | 2.17 | 2.21 | 2.31 |

| b | 1 (baseline) | 1.015 | 1.016 | 1.017 | 1.018 | 1.019 |
|---|---|---|---|---|---|---|
| $T' = 50$ | 4.43 | 2.53 | 2.51 | 2.49 | 2.53 | 2.55 |

Table 18: FID of EDM-ES achieved on CIFAR-10 under different $\lambda(t)$ ($\lambda(t) = b$) and unconditional Heun sampler, b=1 represents EDM.

| b | 1 (baseline) | 1.0005 | 1.0006 | 1.0007 | 1.0008 |
|---|---|---|---|---|---|
| $T' = 35$ | 1.97 | 1.948 | 1.947 | 1.949 | 1.953 |

| b | 1 (baseline) | 1.004 | 1.005 | 1.006 | 1.007 |
|---|---|---|---|---|---|
| $T' = 13$ | 7.16 | 6.60 | 6.55 | 6.54 | 6.55 |

## A.12 EPSILON SCALING ALLEVIATES EXPOSURE BIAS

In Section 5.2, we have explicitly shown that Epsilon Scaling reduces the exposure bias of diffusion models via refining the sampling trajectory and achieves a lower $\delta_t$ on CIFAR-10 dataset.

We now replicate these experiments on other datasets using the same base model ADM and 20-step sampling. Fig. 9 and Fig. 10 display the corresponding results on CIFAR-10 and FFHQ 128×128 datasets. Similar to the phenomenon on LSUN tower 64×64 (Fig. 5 and Fig. 4), Epsilon Scaling consistently obtains a smaller exposure bias $\delta_t$ and pushes the sampling trajectory to the vector field learned in the training stage.

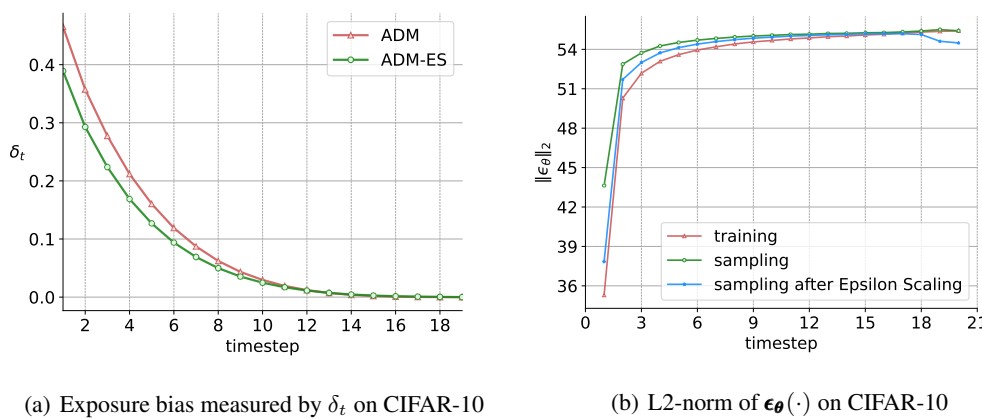

(a) Exposure bias measured by $\delta_t$ on CIFAR-10      (b) L2-norm of $\boldsymbol{\epsilon_\theta}(\cdot)$ on CIFAR-10

Figure 9: Left: Epsilon Scaling achieves a smaller $\delta_t$ at the end of sampling ($t = 1$). Right: after applying Epsilon Scaling, the sampling $\|\boldsymbol{\epsilon_\theta}\|_2$ (blue) gets closer to the training $\|\boldsymbol{\epsilon_\theta}\|_2$ (red)

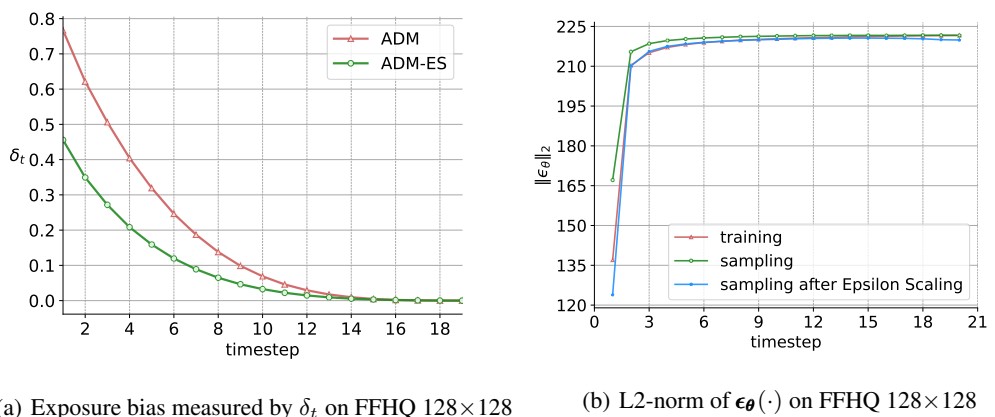

(a) Exposure bias measured by $\delta_t$ on FFHQ 128×128      (b) L2-norm of $\boldsymbol{\epsilon_\theta}(\cdot)$ on FFHQ 128×128

Figure 10: Left: Epsilon Scaling achieves a smaller $\delta_t$ at the end of sampling ($t = 1$). Right: after applying Epsilon Scaling, the sampling $\|\boldsymbol{\epsilon_\theta}\|_2$ (blue) gets closer to the training $\|\boldsymbol{\epsilon_\theta}\|_2$ (red).

## A.13 FID RESULTS ON DIT BASELINE

In addition to UNet (Ronneberger et al., 2015) backbone, we also test Epsilon Scaling on diffusion models using Vision Transformers (Dosovitskiy et al., 2020). Table 19 presents the FID of DiT (Peebles & Xie, 2023) and our DiT-ES on ImageNet 256×256 under uniform $\lambda(t) = b$. Again, Epsilon Scaling achieves consistent sample quality improvement over different sampling steps, indicating that the exposure bias exists in both UNet and Transformer backbones.

Table 19: FID on ImageNet 256×256 using DiT baseline

| Model | $T'$ | | |
|---|---|---|---|
| | 20 | 50 | 100 |
| DiT | 12.95 | 3.71 | 2.57 |
| DiT-ES | **10.00** **(b=0.965)** | **3.30** **(b=0.989)** | **2.52** **(b=0.995)** |

## A.14 QUALITATIVE COMPARISON

In Section 5.6, we have presented the sample quality comparison between the base model sampling and Epsilon Scaling sampling on FFHQ 128×128 dataset. Applying the same experimental settings, we show more qualitative contrasts between ADM and ADM-ES on the dataset CIFAR-10 32×32 (Fig. 11), LSUN tower 64×64 (Fig. 12), ImageNet 64×64 (Fig. 13) and ImageNet 128×128 (Fig. 14). Also, we provide the qualitative comparison between LDM and LDM-ES on the dataset CelebA-HQ 256×256 (Fig. 15). These sample comparisons clearly state that Epsilon Scaling effectively improves the sample quality from various perspectives, including illumination, colour, object coherence, background details and so on.

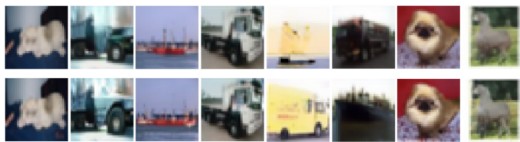

Figure 11: Qualitative comparison between ADM (first row) and ADM-ES (second row) on CIFAR-10 32×32 using 100-step sampling.

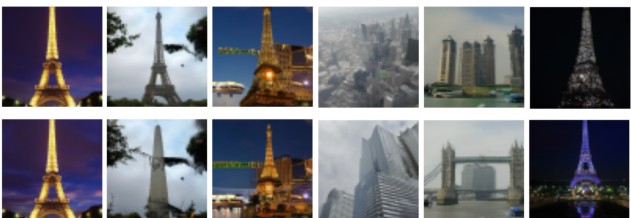

Figure 12: Qualitative comparison between ADM (first row) and ADM-ES (second row) on LSUN tower 64×64 using 100-step sampling.

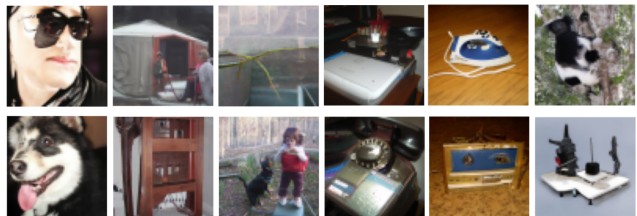

Figure 13: Qualitative comparison between ADM (first row) and ADM-ES (second row) on ImageNet 64×64 using 100-step sampling.

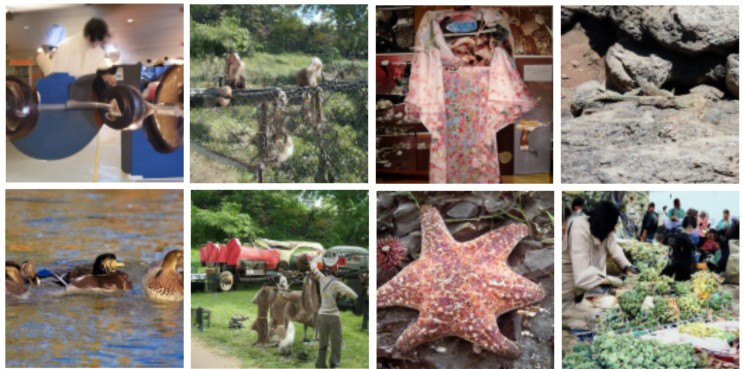

Figure 14: Qualitative comparison between ADM (first row) and ADM-ES (second row) on ImageNet 128×128 using 100-step sampling.

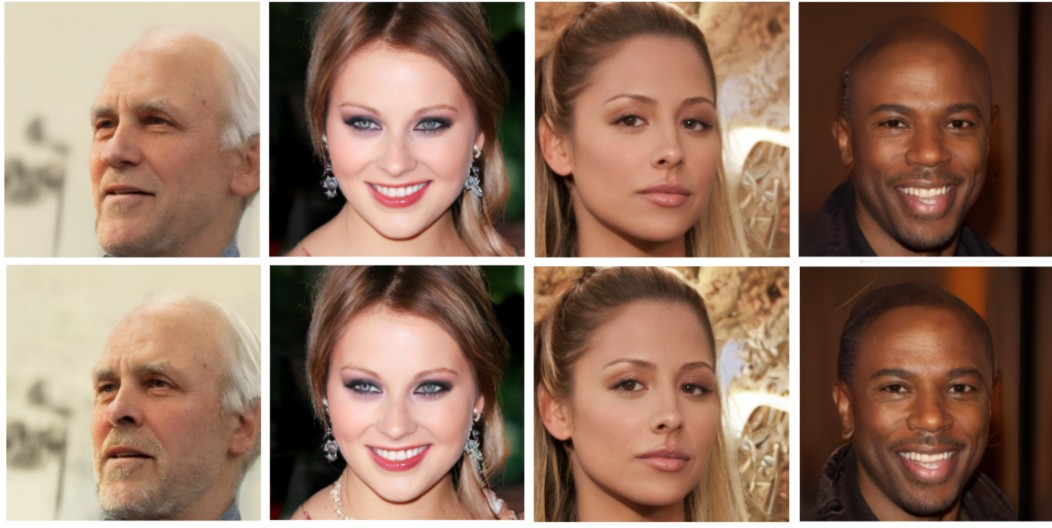

Figure 15: Qualitative comparison between LDM (first row) and LDM-ES (second row) on CelebA-HQ 256×256 using 100-step sampling.

