# OpenReview forum: "Elucidating the Exposure Bias in Diffusion Models"
_ICLR.cc/2024/Conference — ICLR 2024 poster_

### Official Review · Reviewer_tyf7 · 2023-10-31

**Soundness:** 3 good
**Presentation:** 3 good
**Contribution:** 3 good
**Rating:** 8
**Confidence:** 4

**Summary:**

This paper studies the problem of exposure bias in diffusion models.  Exposure bias is defined as the input mismatch between $x_t$ during training and $\hat{x}_t$ during sampling, which results error accumulation during sampling and eventual sampling drift. This in turn affects the quality of generated images.

The primary cause of exposure bias is ascribed to the difference between the growth truth posterior distribution $q(x_{t-1} | x_t, x_0)$ and the sampling distribution $q(x_{t-1} | x_t, \hat{x}^t_\theta)$. In practice, the difference between the prediction $\hat{x}_\theta^t$ and ground truth $x_0$ is non-zero. The paper analytically shows that this is due to increased variance at sampling time which results in accumulation of error during sampling. Inspired by the error in variance, a metric to measure exposure bias is also proposed by drawing inspiration from FID.

To alleviate exposure bias, this work proposes a method called Epsilon scaling which can be used at sampling and does not require re-training/finetuning of diffusion models. The core idea is to scale down the predicted $\epsilon_\theta^s$ at sampling so that it is closer to training $\epsilon_\theta^t$.

Qualitative and quantitative results indicate that the proposed method results in improved FID scores and helps in alignment of sampling trajectory with training trajectory, thus reducing exposure bias.

**Strengths:**

1. The paper is well-written. The paper provides a clear explanation about the problem of exposure bias and its causes by deriving expressions for sampling probability distribution that shows increased variance. It also provides clear intuition for different choices while designing the proposed method. For instance, the intuition for correcting $\epsilon_\theta$ to reduce exposure bias instead of reducing variance error has been explained clearly.
2. The method is relatively simple and easy to implement as it introduces a scaling factor in the sampling process of diffusion models and does not require re-training of diffusion models. Previous method by Ning et al. (2023) mitigates this issue by perturbing inputs during training which requires retraining diffusion models.
3. The proposed method improves FID scores significantly for different families of diffusion models like DDPM, DDIM, ADM, EDM and LDM.
4. The primary contribution of this work which is empirical demonstration of the fact that correction of exposure bias can be done without retraining diffusion models is valuable.

**Weaknesses:**

1. The proposed method of Epsilon scaling is not tuning-free as it is sensitive to the choice of hyper parameter of the schedule $\lambda_t = kt+ b$ and thus requires extensive hyper parameter tuning. The choice of optimal hyperparameters varies for each dataset as well as the number of sampling steps T. As per Table 10, Table 11 and Table 12, values of k and b have been tuned up to 4th or even 5th decimal place. For instance, there are values like k=0.00022 , b=1.00291 in Table 10. This degree of hyperparameter tuning seems a bit extreme.  The paper is also missing relevant sensitivity analysis for hyperparameters k and b, and it would be useful to include it (both over large strides and small strides).
2. The process of finding optimal hyperparameters for Epsilon scaling seems to be slow and can be computationally intensive.  Usually, generating 10K sampling to calculate FID takes several minutes for small sampling steps and several hours for large sampling steps. Tuning hyper-parameters for each dataset and each choice of sampling length T, can thus require lot of GPU compute and time. While the method itself is simple, tuning this method for optimal performance can be tricky.
3. In Table 3, for CIFAR-10, values of EDM for VP and VE should be reported for NFE=35 which gets FID = 1.97 (VP) and 1.98 (VE) on CIFAR-10, respectively. Underreporting FID values of EDM is a bit misleading.

**Questions:**

1. In Figure 5, Figure 9b and Figure 10b, what is the reason for smaller values of $\| \epsilon \|_2$ of ADM-ES compared to its values at training time towards end of sampling (time steps 18-21)?
2. Could the authors include the number of sampling steps in qualitative results in Figures 11-15?
3. Could the authors indicate the amount of compute (number of GPUs as well as time) needed to find optimal hyperparameters?

---

> ### Author Response · Authors · 2023-11-20
>
> Thank you very much for the insightful and helpful suggestions. We answer each question in detail below:
>
> **Q1. The proposed method of Epsilon scaling is not tuning-free as it is sensitive to the choice of hyper parameter of the schedule $\lambda_t = kt+b$ and thus requires extensive hyper parameter tuning. The paper is also missing relevant sensitivity analysis for hyperparameters k and b, and it would be useful to include it (both over large strides and small strides).**
>
> **A1**. Thank you for pointing this out, Reviewer gcKq also asked a similar question. Overall, the parameters k,b are insensitive to the FID result, please refer to our answer A2 to Reviewer gcKq for details. Since the linear schedule $\lambda_t = kt+b$ is determined from its optimal uniform schedule $\lambda_t=b^*$, b of the linear schedule is actually computed rather than brute-force searched. Thus in the linear schedule case, only k is searched with several trials. The sensitivity analysis is of great interest and helpful for the reader to understand the easiness of our method, for example, we present a curve that shows the FID w.r.t the full range of $b$. We decided to include it in the paper later on.
>
>
> **Q2. The process of finding optimal hyperparameters for Epsilon scaling seems to be slow and can be computationally intensive. Usually, generating 10K sampling to calculate FID takes several minutes for small sampling steps and several hours for large sampling steps. Tuning hyper-parameters for each dataset and each choice of sampling length T, can thus require lot of GPU compute and time. While the method itself is simple, tuning this method for optimal performance can be tricky.**
>
> **A2**. As we mentioned above, the determination of $\lambda_t$ is practically easy and not expensive. For each dataset and network, we find the optimal $\lambda_t$ by always using a single A100 GPU, costing from several minutes to three hours. We hope this tackles your concerns about the computational cost of our method.
>
>
> **Q3. In Table 3, for CIFAR-10, values of EDM for VP and VE should be reported for NFE=35 which gets FID = 1.97 (VP) and 1.98 (VE) on CIFAR-10, respectively. Underreporting FID values of EDM is a bit misleading**
>
> **A3**. Thank you for the question, to make sure we measure the true improvement of our method, we re-run the sampling and FID measurement for EDM baseline in the same machine and environment. We report FID=1.97 for EDM (VP, uncond) and FID=1.82 for EDM (VP, cond) in the paper, the latter is different from FID=1.79 reported in EDM, which is possibly due to the machine difference. To avoid misunderstandings, we will add a sidenote in the table to mention the result is reported by re-running the baseline.
>
>
> **Q4. In Figure 5, Figure 9b and Figure 10b, what is the reason for smaller values of $|\epsilon|$ of ADM-ES compared to its values at training time towards end of sampling (time steps 18-21)?**
>
> **A4**. Thank you for the question, timestep 18-21 in these three figures means the beginning of the sampling, which we should clarify in the figure (will update it). The smaller $\epsilon$ norm corresponds to the effect that we push the sampling trajectory away from the original trajectory.
>
>
> **Q5. Could the authors include the number of sampling steps in qualitative results in Figures 11-15?**
>
> **A5**. That is a good suggestion, we will include it in the next version paper.
>
>
> **Q6. Could the authors indicate the amount of compute (number of GPUs as well as time) needed to find optimal hyperparameters?**
>
> **A6**. Finding an optimal uniform $\lambda_t$ takes 6 to 10 trails of searching, which corresponds to several minutes to 3 hours in a single A100 GPU, depending on NFE and dataset resolution. We will include the details in the appendix.

---

> > ### Comment · Reviewer_tyf7 · 2023-11-21
> >
> > Thank you for your responses to my questions.
> >
> > For Q1. Could you please indicate the values of $k$ in your tables for hyperparameter search? Currently, the response includes only values of $b$ in Tables 11 and 12 above. I would also argue that the current range of $b$ is narrow, and therefore I don't completely agree with the statement "FID gain can be achieved in a wide range of k,b, which indicates the insensitivity of the hyperparameters."
> >
> > I am also curious about the response to Q4 raised by Reviewer FqVz. I have calibrated my current rating on the basis of rebuttal and responses to other reviews, but I am open to updating it in the future.

---

> > > ### Author Response · Authors · 2023-11-21
> > >
> > > Thank you for the response, for a better clarification of our k selection, we elaborate the complete procedure and full search range for k in a linear $\lambda_t$ case. Taking the LSUN 64x64 as an example, we found that the optimal uniform $\lambda_t^*=1.011$ for NFE=20, then to decide the linear schedule $\lambda_t=kt+b$, we search the optimal $k$  on a stride $0.0002$ and b is calculated by retaining $\sum \lambda_t = \sum \lambda_t^*$, the full range of $k$ we implemented is shown in Table 13.  More experiments on FFHQ128x128, ImageNet 64 are presented in Table 14, 15. Overall, we highlight the ease and low cost of finding the schedule $\lambda_t$. Although different opinions might exist regarding the definitionof sensitivity (how much is sensitive, how much is insensitive), the goal of the sensitivity analysis or people’s expectation is that the schedule $\lambda_t$ is easy to find in practice, which we have shown.
> > >
> > > Table 13. FID with linear schedule $\lambda_t=kt+b$ on LSUN 64x64 dataset under NFE=20
> > >
> > > | k | 0 (uniform) | 0.0002 | 0.0004 | 0.0006 | 0.0008 | 0.0010 |
> > > | --- | --- | --- | --- | --- | --- | --- |
> > > | ADM-ES | 8.22 | 8.10 | 8.01 | 7.75 | 7.60 | 7.69 |
> > >
> > > Table 14. FID with linear schedule $\lambda_t=kt+b$ on FFHQ 128x128 dataset under NFE=20
> > >
> > > | k | 0 (uniform) | 0.0003 | 0.0004 | 0.0005 | 0.0006 | 0.0007 | 0.0008 |
> > > | --- | --- | --- | --- | --- | --- | --- | --- |
> > > | ADM-ES | 26.14 | 24.50 | 24.18 | 24.35 | 24.41 | 24.73 | 24.83 |
> > >
> > > Table 15. FID with linear schedule $\lambda_t=kt+b$ on ImageNet 64x64 dataset under NFE=20 (FID computed by 10k samples)
> > >
> > > | k | 0 (uniform) | 0.0001 | 0.0002 | 0.0003 | 0.0004 | 0.0005 |
> > > | --- | --- | --- | --- | --- | --- | --- |
> > > | ADM-ES | 10.23 | 10.19 | 9.98 | 10.22 | 10.27 | 10.31 |

---

> > > > ### Comment · Reviewer_tyf7 · 2023-11-23
> > > >
> > > > Thank you for sharing these additional results. I do think that this paper offers a nice perspective on correcting exposure bias and  empirical results are encouraging, as the gains seem consistent, even though they are minor in some cases like DDIM. One of my major concerns was sensitivity of this method to hyperparameter. This has been addressed to some extent. I still feel that the need for 6-8 trials and upto 3 hours GPU time for finetuning is a lot, given that these hyperparameters don't seem to transfer across timesteps, datasets or specific parametrization of diffusion process. Overall, I do think that this paper has some interesting insights and therefore I'm increasing my score.

---

> > > > > ### Author Response · Authors · 2023-11-23
> > > > >
> > > > > Dear reviewer, thank you very much for the reply and for being active during the whole discussion!

---

### Official Review · Reviewer_gcKq · 2023-11-01

**Soundness:** 3 good
**Presentation:** 3 good
**Contribution:** 2 fair
**Rating:** 6
**Confidence:** 2

**Summary:**

This work focuses on the `exposure` bias problem for diffusion models. The exposure bias is caused by error accumulation across the sampling trajectory since during inference only samples from the previous timestep step are used, while during training samples are exposed to ground truth training. The work then quantifies and analyzes the exposure bias during inference, during different timesteps showing the error is high by the end of sampling (for multi-step sampling) and shows a monotonic trend. Lastly, the paper proposes epsilon scaling as an inference time sampling strategy to mitigate this issue and shows higher quality generations compared to other samplers while (slightly) improving FID.

**Strengths:**

1. The work does an overall good job of introducing and explaining exposure bias. The section itself was didactic and a valuable portion of the paper. Results are shown for simple single and multi-step samplings that quantify the variance error caused by exposure bias.
2. The work then proposes a simple approach to reduce the exposure bias, by scaling the norm of the predicted noise term. The work also shows that the scaling factor needs to be handled differently when sampling for different number of steps.
3. Results are shown for several different variants of diffusion models, and several baseline samplers are considered. The proposed method shows competitive FID scores with similar or fewer timesteps.

**Weaknesses:**

1. Results for the proposed epsilon-scaling mechanism with DDIM solver aren't impressive. There's only a marginal change in the FID score, compared to the DDIM sampler which is used as a popular framework for many diffusion models. Here, the gains are marginal even for cases with reasonable timesteps (~50 or more).
2. There is little information provided regarding how the scaling factor $k$ and $b$ are selected, and how many hyperparameters were searched for optimal $k$ and $b$. It would also be good to quantify sensitivity to these hyperparameters.

**Questions:**

1. Does the method perform worse than Heun solvers with suboptimal $k$ and $b$ selection?

---

> ### Author Response · Authors · 2023-11-20
>
> Thank you for your reviews and constructive suggestions. Our relies to each question is listed below.
>
> **Q1. Results for the proposed epsilon-scaling mechanism with DDIM solver aren't impressive. There's only a marginal change in the FID score, compared to the DDIM sampler which is used as a popular framework for many diffusion models. Here, the gains are marginal even for cases with reasonable timesteps (~50 or more)**
>
> **A1**. Although the FID gain of Epsilon Scaling in DDIM is less than the gain in DDPM, our method is still useful in DDIM sampler as a simple plug-in sampling solution. Except for the outlier in CelebA 20-step sampling, the FID improvement is also non-trivial and can be significant from 4.06 to 3.38 in CIFAR-10. In the case that DDIM-ES gains little benefit, one can easily get a better sample quality by increasing the value of $\eta$ from 0 to 0.2, 0.5, etc.
>
>
> **Q2. There is little information provided regarding how the scaling factor $k$
>  and $b$ are selected, and how many hyperparameters were searched for optimal $k$ and $b$. It would also be good to quantify sensitivity to these hyperparameters.**
>
> **A2**.  Thank you for the suggestions. First, the FID gain can be achieved in a wide range of k,b, which indicates the insensitivity of the hyperparameters. For example, in Tables 11, 12, we present the FID and parameter b across ADM and EDM baselines. We are going to add a more complete sensitivity graph and analysis in the paper.
>
> Table 11. FID on ADM baseline and CIFAR-10 under different parameters b (b=1 represents the baseline)
> | b | 1 (baseline) | 1.015 | 1.016 | 1.017 | 1.018 | 1.019 |
> | --- | --- | --- | --- | --- | --- | --- |
> | NFE=100 | 3.37 | 2.20 | 2.18 | 2.17 | 2.21 | 2.31 |
> | NFE=50 | 4.43 | 2.53 | 2.51 | 2.49 | 2.53 | 2.55 |
>
> Table 12. FID on EDM baseline and CIFAR-10 under different parameters b (b=1 represents the baseline)
> | b | 1 (baseline) | 1.0005 | 1.0006 | 1.0007 | 1.0008 |
> | --- | --- | --- | --- | --- | --- |
> | NFE=35 | 1.97 | 1.948 | 1.947 | 1.949 | 1.953 |
> |  |  |  |  |  |  |
> | b | 1 (baseline) | 1.004 | 1.005 | 1.006 | 1.007 |
> | NFE=13 | 7.16 | 6.60 | 6.55 | 6.54 | 6.55 |
>
> Thanks to the insensitivity of parameters $k$, $b$, the parameter searching is effortless, searching for a constant $b$ takes no more than 10 trials for a specific network, dataset and sampling step. We usually take 2 rounds of searching, first doing the coarse search with a stride of 0.005, then doing a fine search with a stride of 0.001. When using linear schedule $\lambda_t=kt+b$, we first find its optimal uniform schedule $\lambda_t=b^*$, then search for the optimal slope $k$ while maintaining $\sum\lambda_t$ unchanged (i.e. $\sum\lambda_t=NFE \times b^* $), in this way, $b$ of the linear schedule is calculated, we got only $k$ is searched. There are already descriptions of the selection of $k,b$ in Appendix A.8, but we will accordingly add more details to the appendix.
>
>
> **Q3. Does the method perform worse than Heun solvers with suboptimal $k$ and $b$ selection?**
>
> **A3**. Due to the wide range of near-optimal solutions, our method still works well under suboptimal k and b, see Table 12 where we use Heun solver.

---

### Official Review · Reviewer_ujfx · 2023-11-04

**Soundness:** 3 good
**Presentation:** 2 fair
**Contribution:** 3 good
**Rating:** 6
**Confidence:** 4

**Summary:**

This paper investigates the exposure bias problem in diffusion models by modeling sampling distribution, based on which they attribute the prediction error at each sampling step as the root cause of the exposure bias issue. They discuss potential solutions to this issue and propose a metric for it. Along with the elucidation of exposure bias, we propose a simple, yet effective, training-free method called Epsilon Scaling to alleviate the exposure bias.
Experiments on various diffusion frameworks, unconditional and conditional settings, and deterministic vs. stochastic sampling verify the effectiveness of this method.

**Strengths:**

1. It is interesting and insightful to take in-depth exploration on the exposure bias problem in diffusion models. This paper connects exposure bias with prediction error and gives the expressions of prediction error.

2. To solve the exposure bias in a learning-free manner, this paper proposes to scale the noise prediction in the sampling process to match the noise prediction in the training process.

3. Solid experiments. Extensive experiments demonstrate the generality of Epsilon Scaling and its applicability to different diffusion architectures.

**Weaknesses:**

1. The main concern lies in the assumption at the start of the derivation. In part 3.2, the authors assume that the reconstructed image x_{\theta}^t at the sampling process follows the Gaussian distribution, where the mean is the GT image x_0, and the variance is the Gaussian noise. This conflicts with some intuitive observations. For example, the reconstructed image x_{\theta}^t is often a degraded version of the GT image, and the mean of x_{\theta}^t is different from x_0.

2. Some formulations, notations and explanations in part 3.1 and 3.1 are redundant. It is highly recommended to improve the organization and the writing of this section.

3. It is necessary to discuss the further meaning and defects of exposure bias correction. For instance, the necessity of matching the forward and sampling process. Reducing such bias forces the generated images to be more similar to the training images in distribution, i.e, more data repetition in the generated images. The authors are encouraged to give more discussions.

**Questions:**

Please refer to the questions in the Weaknesses.

---

> ### Author Response · Authors · 2023-11-20
>
> Thank you very much for your helpful comments and we now answer each of your questions.
>
> **Q1. The main concern lies in the assumption at the start of the derivation. In part 3.2, the authors assume that the reconstructed image x_{\theta}^t at the sampling process follows the Gaussian distribution, where the mean is the GT image x_0, and the variance is the Gaussian noise. This conflicts with some intuitive observations. For example, the reconstructed image x_{\theta}^t is often a degraded version of the GT image, and the mean of x_{\theta}^t is different from x_0.**
>
> **A1**. The generated sample is the result of multi-step sampling, where the Gaussian approximation of $x_0$ prediction error holds in a single step (please see our answer A1 to reviewer nv6a). For multi-step sampling, this Gaussian might be loose and exhibit the mean drift.
>
>
> **Q2. Some formulations, notations and explanations in part 3.1 and 3.1 are redundant. It is highly recommended to improve the organization and the writing of this section.**
>
> **A2**. Thanks for your suggestions, we will simplify section 3.1 since the DDPM formulation is already well-known in the field.
>
>
> **Q3. It is necessary to discuss the further meaning and defects of exposure bias correction. For instance, the necessity of matching the forward and sampling process. Reducing such bias forces the generated images to be more similar to the training images in distribution, i.e, more data repetition in the generated images. The authors are encouraged to give more discussions.**
>
> **A3**. From the sampling trajectory perspective, exposure bias correction indeed tries to match the training and sampling trace. The main benefit of this match is to avoid the intermediate sample $\hat{x_t}$ falling into the problematic vector field that has never been trained in the training phase. From the sample quality perspective, the exposure bias correction corresponds to the correction of overexposure, underexposure, and detail defects.
>
> We empirically observed that the generated image samples before and after applying Epsilon Scaling share the same semantic mode. Thus, the exposure bias correction does not affect the diversity of the synthesised samples. Our experiments of recall further show that the correction of exposure bias results in even slightly better sample diversity (see Table 10 below). Thus, reducing exposure bias does not lead to data repetition. We plan to add this discussion in section 5.6
>
>
> Table 10. Recall of ADM and ADM-ES using 100-step sampling
>
> |  | CIFAR-10 | LSUN 64x64 | FFHQ 128x128 | ImageNet 64x64 | ImageNet 128x128 |
> | --- | --- | --- | --- | --- | --- |
> | ADM | 0.591 | 0.605 | 0.497 | 0.621 | 0.586 |
> | ADM-ES | 0.613 | 0.606 | 0.545 | 0.632 | 0.592 |

---

### Official Review · Reviewer_FqVz · 2023-11-04

**Soundness:** 2 fair
**Presentation:** 2 fair
**Contribution:** 3 good
**Rating:** 6
**Confidence:** 4

**Summary:**

The paper studies the "exposure bias" in diffusion models, which boils down to the statistical discrepancy between neural predictions at different time $t$. To alleviate the issue, the authors propose a simple scaling strategy to match the norm of neural predictions at training and sampling time. The idea is to calculate the related empirical statistics and then determine the scaling factor at different time $t$. Empirically, the proposed scaling method improve the baseline across architectures, noisy schedule and datasets.

**Strengths:**

- The paper tries to address an important issue in diffusion models, where the initial error accumulation can negatively affect the quality of the generated samples.

- It introduces a method that employs the empirical $\ell_2$ ratio during both training and sampling phases to decide the appropriate scaling factor. This technique is straightforward yet proves to be effective.

- Extensive experiments show that the proposed method can consistently improve the pre-trained models across datasets.

**Weaknesses:**

- Several prior works have observed and identified the "exposure bias" problem studied in the current paper. It would be helpful to discuss them in the paper: (1) Section 4 (practical considerations) and Fig 13 in EDM [1] points out that the neural network tends to remove slightly too much noise. Hence they use an inflated noise to counteract it. I think adding $S_{noise}$ into ODE or SDE samplers is a valid baseline for the current paper. (2) Section 4.2 in PFGM [2] / Section 5, Fig 4.b in PFGM++ [3] further dig into the exposure bias problem and show that the strong norm-t relation in the diffusion model is the cause.

- The current paper demonstrates the discrepancy of the prediction norms (Fig 2) without further examining the reason. [2] seems to offer a plausible explanation for this phenomenon. If their argument is true, this "exposure bias" mainly occurs in large $t$ (norm-t is strongly correlated). I guess one simple experiment to validate the hypothesis is to use true score up to a certain time $t$ during sampling and study the effects (like current Fig.2, but the green and red curve would be the same in [t,T]).


- It's encouraging to see that the method improves over EDM, especially in the small NFE regime. Could the authors try to apply the method on PFGM++, which is claimed to be more robust to the "exposure bias"? It could also verify if or not the exposure bias is due to the strong norm-t correlation pertaining to diffusion models.

### Minors

- I think the prediction error is not $x^t_\theta - x_0$ but $x^t_\theta - E_{x_0|x_t}[x_0]$.

[1] Elucidating the Design Space of Diffusion-Based Generative Models, NeurIPS 2022

[2] Poisson Flow Generative Models, NeurIPS 2022

[3] PFGM++: Unlocking the Potential of Physics-Inspired Generative Models, ICML 2023

Overall I'm impressed by the empirical performance of the proposed methods. However, there are many loose threads (the reasoning of exposure bias, the question below, comparing to related work (e.g., EDM inflating). I will raise my score if the authors address some of them.

## Update

I would like to thank the authors for providing the extra experiments. I raised my score to 6.

For Q4, sorry I made some mistakes in my previous question. What I mean is that if we deliberately scale the network prediction $\epsilon_\theta$ by $\lambda_t$ (i.e., the network prediction error studied in the paper), the resulting solution seems not related to the $\Delta N(t)=\int_t^T \lambda_t$. Concretely, the new ODE is $dx_t /dt = \lambda_t /t x_t $, and the solution is related to the time integration of $t\lambda_t$. So the argument in the paper appears to be inconsistent with this simple example.

**Questions:**

- I don't immediately see why the quotient of $\Delta N(t) = |\epsilon^s|/|\epsilon^t|$ can be translated into the scaling factor at $\epsilon(x_t,t)$ during sampling. Because $\Delta N(t) = |\epsilon^s|/|\epsilon^t|$ and $\Delta N(t)=\int_t^T \lambda_t$ only implies that $\frac{d |\epsilon^s(x_t,t)|/|\epsilon^t(x_t,t)|}{dt}=\lambda_t$.

I tried the simplest data distribution I can come up with --- $p(x)=\delta(x-0)$. The argument even failed in this case. (I use the notation in EDM and assume $t=\sigma$). The true $\epsilon$ at $x_t$ is $\epsilon(x_t,t) = -\frac{x_t}{t^2}$. By solving the diffusion ODE $dx/dt = -\frac{x_t}{t}$ we get $x_t = \frac{Tx_T}{t}$ (assume the start point at time $T$ is $x_T$). However, when we scale $\epsilon$ by $\lambda_t$, the resulting $x_t$ does not directly relate to $\Delta N(t)=\int_t^T \lambda_t$, but a interval over $\lambda_t/t$. I suggest rethinking the principled way of scaling by working out the toy example first.

---

> ### Author Response · Authors · 2023-11-20
>
> Thank you very much for the interesting and insightful comments, our answers to each question are shown below:
>
>
> **Q1. Several prior works have observed and identified the "exposure bias" problem studied in the current paper. It would be helpful to discuss them in the paper: (1) Section 4 (practical considerations) and Fig 13 in EDM [1] points out that the neural network tends to remove slightly too much noise. Hence they use an inflated noise to counteract it. I think adding $S_{noise}$ into ODE or SDE samplers is a valid baseline for the current paper. (2) Section 4.2 in PFGM [2] / Section 5, Fig 4.b in PFGM++ [3] further dig into the exposure bias problem and show that the strong norm-t relation in the diffusion model is the cause.**
>
> **A1**. We think our Epsilon Scaling is consistent with EDM’s statement: the neural network tends to remove slightly too much noise, because our solution scales down the predicted noise, thus avoiding removing too much noise. We will update the SDE sampler experiments once we get the results. As for PFGM++ discussion, please refer to our answers A2 and A3.
>
>
> **Q2. The current paper demonstrates the discrepancy of the prediction norms (Fig 2) without further examining the reason. [2] seems to offer a plausible explanation for this phenomenon. If their argument is true, this "exposure bias" mainly occurs in large $t$ (norm-t is strongly correlated). I guess one simple experiment to validate the hypothesis is to use true score up to a certain time
>  during sampling and study the effects (like current Fig.2, but the green and red curve would be the same in [t,T]).**
>
> **A2**. First of all, we reckon the strong norm-t opinion does not contradict our claim that the prediction error of $x_0$ is the cause of exposure bias. We are convinced that PFGM++ is robust to prediction error compared with diffusion models. So they are two concepts and can be concurrent. Also, we argue that exposure bias exists in the whole sampling chain of diffusion models. The experiment in paper [1] supports our opinion, where they measure the FID by leaking the ground truth $x_t$, $t \sim(1, T)$ and $T=1000$, to the network at the beginning of sampling, then the sampling starts at $x_t$. We replicate their results on ADM and ImageNet 32 in Table 6. We see that the FID is still far from near-zero even though the sampling starts at $x_{300}$ and $x_{100}$ which jump over the large t area (norm-t is strongly correlated).
>
> Table 6. FID on ImageNet 32
> | t | 100 | 300 | 500 | 700 | 1000 |
> | --- | --- | --- | --- | --- | --- |
> | ADM | 0.98 | 1.81 | 2.59 | 3.11 | 3.54 |
>
> [1] Mang Ning, Enver Sangineto, Angelo Porrello, Simone Calderara, and Rita Cucchiara. Input perturbation reduces exposure bias in diffusion models. ICML, 2023
>
>
> **Q3. It's encouraging to see that the method improves over EDM, especially in the small NFE regime. Could the authors try to apply the method on PFGM++, which is claimed to be more robust to the "exposure bias"? It could also verify if or not the exposure bias is due to the strong norm-t correlation pertaining to diffusion models.**
>
> **A3**. We think PFGM++ is an interesting baseline to apply our method. We present the results on PFGM++ (D=128) and PFGM++ (D=2048) under unconditional sampling (see Table 7, 8). We found that Epsilon Scaling (ES) is compatible with PFGM++ and even enjoys the extra benefit: under very small NEF regimes, the FID gain of ES is still significant. We believe the robustness to prediction error of PFGM++ is the reason. We would like to include the full experiments on PFGM++ in the main paper and discuss the strong norm-t relation in the paper as well.
>
>  Table 7. FID on PFGM++ baseline (D=128) and CIFAR-10
> | NFE | 9 | 13 | 21 | 35 |
> | --- | --- | --- | --- | --- |
> | PFGM++ | 37.82 | 7.55 | 2.34 | 1.92 |
> | PFGM++ with ES  | 17.91 | 4.51 | 2.31 | 1.91 |
>
> Table 8. FID on PFGM++ baseline (D=2048) and CIFAR-10
> | NFE | 9 | 13 | 21 | 35 |
> | --- | --- | --- | --- | --- |
> | PFGM++ | 37.16 | 7.34 | 2.31 | 1.91 |
> | PFGM++ with ES  | 17.27 | 4.88 | 2.20 | 1.90 |
>
> Regarding the small NFE regime on EDM baseline, we present the results of using the Euler solver below, since we found that, when NFE is smaller than 10 in EDM, the Heun solver performs much worse than the Euler solver mainly due to the twice-larger stepsize and severe truncation error.
>
> Table 9. FID on EDM baseline under small NFE regimes (CIFAR-10 and unconditional)
> | NFE | 5 | 7 | 9 | 13 |
> | --- | --- | --- | --- | --- |
> | EDM | 68.69 | 36.58 | 22.97 | 12.28 |
> | EDM-ES | 55.26 | 27.29 | 16.34 | 8.39 |
>
>
> **Q4. I don't immediately see why the quotient of $\Delta N(t)=|\epsilon_s| / |\epsilon_t|$ can be translated into the scaling factor at $\epsilon (x_t, t)$ during sampling. Because $\Delta N(t)=|\epsilon_s| / |\epsilon_t|$ and $\Delta N(t) = \int_{t}^{T} \lambda_t $ only implies that $ \frac{d|\epsilon^s (x_t, t)| / |\epsilon^t (x_t, t)|}{dt} = \lambda_t $.**
>
> **A4**. Updating soon

---

> > ### Author Response · Authors · 2023-11-21
> >
> > Dear reviewer FqVz, Thanks for your question **Q4**, we have some doubts on the case you provided. If we are not mistaken, the forward ODE you defined is $x_t = -t^2 \epsilon$ where $\epsilon \sim N(0, I)$, according to the probability ODE $dx = -\dot\sigma(t) \sigma(t) \nabla_x log p(x) dt$ (equation 1 in EDM paper), you get the ODE $dx/dt = -x_t/t$, soving this ODE, you have  $x_t = \frac{T}{t}x_T$, based on which, we found $x_0$ is infinite. In a similar way, we tried with a similar forword ODE $x_t = t \epsilon$ where $\epsilon\sim N(0, I)$, we derive the reverse ODE $dx/dt =x_t/t$, soving it out leads us to $x_t = \frac{t}{T}x_T$, we now have $x_0=0$. May we ask is this the case you actually mean? Also, it would be appreciated if you could elaborate the details of "when we scale $\epsilon$ by $\lambda_t$, the resulting $x_t $ relates to $\Delta N(t)= \int \lambda_t/t dt$", just to make sure we are on the same page for further discussions. Thank you!

---

> ### Author Response · Authors · 2023-11-23
> **A short summary**
>
> Dear reviewer, we summarize our points of view below:
> 1. Our Epsilon Scaling is consistent with EDM’s statement: the neural network tends to remove slightly too much noise. Our solution scales down the predicted noise thus avoiding removing too much noise.
> 2. The reasoning about the exposure bias is fully analysed throughout Section 3 in our paper.
> 3. We have shown that exposure bias exists in the whole chain of sampling, rather than mainly in the large t area where norm-t is strongly correlated.
> 4. We agree that PFGM++ framework is robust to prediction error, but our argument "the prediction error of $x_0$ is the cause of exposure bias" does not contradict PFGM's opinion. Also, our experiments have shown the existence of exposure bias in PFGM++ and our method can consistently improve it. Therefore, according to your hypothesis, we have verified that the exposure bias is not due to the strong norm-t correlation.
> 5. The $\Delta N(t)$ question is unclear to us for two reasons. First, the solution of your reverse ODE is $x_t = \frac{T}{t}x_T$, this leads to $x_0 = \infty$ which is contradictory with your defined forward diffusion $x_t=-t^2\epsilon$ because $x_0=0$ in this case. Second, the derivation of $\Delta N(t)=\int \lambda_t/t dt$ looks obscure to us and we had tried to get your clarification but failed.
>
> Overall, the questions you raised were very interesting and insightful, but we deeply regret that your zero engagement in the discussion provided us with no opportunity for effective communication. Finally, we are always glad and open to having future discussions after the end of this discussion period. Thanks!

---

### Official Review · Reviewer_nv6a · 2023-11-10

**Soundness:** 4 excellent
**Presentation:** 4 excellent
**Contribution:** 3 good
**Rating:** 6
**Confidence:** 4

**Summary:**

This paper take account into the exposure bias problem in the diffusion model, which is the distribution shift between the distribution derived by the forward diffusion process and the (learned) reverse diffusion process at the same time. The exposure bias phenomenon starts with the observation that the expected noise (or signal) from a noisy input, or a drawn sample within the trajectory, is not accurately evaluated. Precisely, the variance of the expected signal given the more-noisy signal is greater in the generative reverse process than in the forward process. The intuitive approach to overcome this is to directly downscale the noise variance term with respect to the variance ratio, but from existing works, this is infeasible because when doing this, the output variance term will be outbounded to the (theoretically available) noise schedule. However, these existing works does not explain the ill performance in the low-NFE regime. This leads to the hypothesis that the negative effect of the exposure bias exceeds the gain of the optimal variance.

Even though there exist some works that deal with the exposure bias, they all consider re-training of the existing diffusion models, which is costly and sometimes require additional hyperparameter tuning. To alleviate this, this paper proposes a new calibration method called epsilon scaling. The method of epsilon scaling is derived simply by a data-driven approach: draw expected noise from the training set (noisy data as input) and from sampling trajectory (intermediate particle in the trajectory) and compare the expected noise term. The motivating experiment shows that the noise term of the sampling data is always greater than that of the training data, and can be intuitively scaled via the ratio between the expectation of the training and sampling estimated noise mean. The experimental session shows that in a mid-NFE regime (20~100), this shows superior performance compared to non-calibrated cases. Moreover, the paper validates that this method alleviates the exposure bias.

**Strengths:**

* The background and the motivating section is well clarified, by first notifying the necessity of calibrating the expected noise term in the diffusion sampling, and then compare the existing method to the newly proposed method.
* To the best of our knowledge, this paper is the first training-free method that calibrates the distribution drift (e.g. exposure bias), by modifying the neural network output with some dataset statistics.
* The experimental section showed that this training-free method works as a pipeline over a variety of existing diffusion model methods, including the early DDIM/ADM to the more recent LDM/EDM.

**Weaknesses:**

* Although this method is introduced as a simulation-free method, it is not completely simulation-free; in order to determine $\lambda_t$ for each timestep, one can compute the dataset statistics with respect to all intermediate trajectory particle, which require some simulation. (But not heavy.)
 * The assumption that the model value $x_\theta^t$ is averaged to the true $x_0$ should be more verified.
 * In the experimental section, only constant or linearized values are used as the scaling schedule $\lambda_t$. This implies that constant reduction of the expected noise is helpful for the sampling process.
 * Analytic-DPM is also a simulation-free (only calculates the optimal variance over the timesteps) This paper proposed that the limit of the Analytic-DPM method is that this is not advantageous in the low-NFE regime, but this paper reproduces well even in low-NFE regime (NFE=10), if the variance clipping in high-SNR (last or second last sampling step) timesteps is held. The fair comparison in various NFE regime should be done, which can affect the scoring of this paper.

**Questions:**

* What about the case that the NFE is less than 20 or greater than 100?
 * At least for small NFE, I recommend adding some ablation studies on fully-searched $\lambda_t$ with respect to all timesteps, rather than fixing this to the fixed value $b$ or taking a linear approximation.
 * It will more support the method, if the bias of the expected noise $\epsilon_\theta^s$ or $\epsilon_\theta^t$. This helps the reader to understand that by simply scaling down the expected noise calibrates the sampling steps, without taking bias (i.e. translating) the noise.

======

Miscellaneous
 * In the line below Equation 7, does $q ( x_t|x_{t+1},x_\theta^{t+1})$ have the same distribution as $q(x_{t+1}|x_t,x_\theta^t)$?
 * The exposure bias term $\delta_t$ should be more precisely denoted in the main section. This proposed metric is only introduced in detail in the appendix section.
 * It will be helpful if the Analytic-DPM and the existing exposure bias methods are also considered as the benchmark, even though this does not use the baseline reverse diffusion process by modifying
 * It will be also helpful also to be mentioned if the Analytic-DPM requires the heavy clipping of the noise variance $\beta_t$ in the near-signal part of the sampling, which causes

**Details Of Ethics Concerns:**

None.

---

> ### Author Response · Authors · 2023-11-20
>
> Thank you very much for the reviews and helpful comments, we now reply to your questions:
>
> **Q1. The assumption that the model value $x_{\theta}^t$ is averaged to the true $x_0$ should be more verified.**
>
> **A1**. The Gaussian assumption of the single step $x_0$ prediction error was first proposed at Analytic-DPM [1] and later on verified at [2] in their section 5.3 and appendix A.5, where they use Shapiro–Wilk statistical test to verify that $x_{\theta}^t-x_0$ is nearly an isotropic Gaussian, we will also mention this in the paper.
>
> [1] Fan Bao, Chongxuan Li, Jun Zhu, and Bo Zhang. Analytic-dpm: an analytic estimate of the optimal reverse variance in diffusion probabilistic models. ICLR, 2022b
>
> [2] Mang Ning, Enver Sangineto, Angelo Porrello, Simone Calderara, and Rita Cucchiara. Input perturbation reduces exposure bias in diffusion models. ICML, 2023
>
>
> **Q2. Analytic-DPM is also a simulation-free (only calculates the optimal variance over the timesteps) This paper proposed that the limit of the Analytic-DPM method is that this is not advantageous in the low-NFE regime, but this paper reproduces well even in low-NFE regime (NFE=10), if the variance clipping in high-SNR (last or second last sampling step) timesteps is held. The fair comparison in various NFE regime should be done, which can affect the scoring of this paper..**
>
> **A2**. Our observation for Analytic-DPM is that their optimal sampling schedule $\beta_{t}^*$ underperforms the DDPM baseline in the large NFE regime (400~1000), and $\beta_{t}^*$ indeed performs well in the mid and low NFE range. We appreciate your suggestions of verifying our method in various NFE regimes and we add the results in A3
>
>
> **Q3. What about the case that the NFE is less than 20 or greater than 100?**
>
> **A3**. Thanks for mentioning this, our method also works well when NFE<20 and NFE>100. For example, in the ADM baseline, our ADM-ES consistently gains FID improvements on different datasets using NFE=10 and constant Epsilon Scaling (see Table 1). We will roll it out to all baselines we used in our paper after the short discussion period.
>
> Table1. FID under NFE=10 on ADM baseline
> |  | CIFAR-10 | LSUN 64x64 | ImageNet 64x64 (cond) |
> | --- | --- | --- | --- |
> | ADM | 23.37 | 45.35 | 27.90 |
> | ADM-ES (ours) | 16.89 | 23.66 | 20.94 |
>
> As for large NFE regions, we show the full results on Analytic-DDPM baseline (Table 2) and ADM baseline (Table 3), where we use constant $\lambda_t$ for Epsilon Scaling (ES). However, we highlight that it is unnecessary to use very high NFE in practice since NFE=100 achieves the near-optimal FID in most cases while being computationally efficient. Many works in the literature have also noticed this phenomenon [3][4][5].
>
> Table 2. FID on CIFAR-10 using Analytic-DDPM baseline (linear noise schedule)
> | NFE | 20 | 50 | 100 | 200 | 400 | 1000 |
> | --- | --- | --- | --- | --- | --- | --- |
> | Analytic-DDPM | 14.61 | 7.25 | 5.40 | 4.01 | 3.62 | 4.03 |
> | Analytic-DDPM with ES | 11.02 | 5.03 | 4.09 | 3.39 | 3.14 | 3.42 |
>
>
> Table 3. FID on CIFAR-10 using ADM baseline
> | NFE | 200 | 300 | 1000 |
> | --- | --- | --- | --- |
> | ADM | 3.04 | 2.95 | 3.01 |
> | ADM-ES | 2.15 | 2.14 | 2.21 |
>
>
> [3] Nichol, A. Q. and Dhariwal, P. Improved denoising diffusion probabilistic models. In ICML, 2021
>
> [5] Lu, C., Zhou, Y., Bao, F., Chen, J., Li, C., & Zhu, J. Dpm-solver: A fast ode solver for diffusion probabilistic model sampling in around 10 steps. NeurIPS, 2022
>
> [4] Yilun Xu, Mingyang Deng, Xiang Cheng, Yonglong Tian, Ziming Liu, and Tommi Jaakkola. Restart sampling for improving generative processes. NeurIPS, 2023
>
>
> **Q4. At least for small NFE, I recommend adding some ablation studies on fully-searched $\lambda_t$ with respect to all timesteps, rather than fixing this to the fixed value $b$ or taking a linear approximation**
>
> **A4**. Fully searching $\lambda_t$ for each timestep is an expensive ablation study even for small NFE as we need either multi-round simulations for each timestep or exponential effort of brute-force search. Due to the short discussion time, we found it difficult to present the ablation results now, but we will add these to the paper in the future.

---

> ### Author Response · Authors · 2023-11-20
>
> **Q5. It will more support the method, if the bias of the expected noise $\epsilon_{\theta}^s$ or $\epsilon_{\theta}^t$. This helps the reader to understand that by simply scaling down the expected noise calibrates the sampling steps, without taking bias (i.e. translating) the noise**
>
> **It will be also helpful also to be mentioned if the Analytic-DPM requires the heavy clipping of the noise variance $\beta_t$ in the near-signal part of the sampling, which causes**
>
> **A5**. It looks like these two questions are sort of incomplete, would you mind re-stating your questions?
>
>
> **Q6. In the line below Equation 7, does $q(x_t | x_{t+1}, x_{\theta}^{t+1})$ have the same distribution as $q(x_{t+1} | x_t, x_{\theta}^t)$ ?**
>
> **A6**. Thank you for pointing this out, we intended to say that $q(\hat{x_t}| x_{t+1}, x_{\theta}^{t+1})$ and $q(x_{t-1}| x_{t}, x_{\theta}^{t})$ are in the same function format and they are just the difference in notation. Clear clarification would be made, such as, 'we now compute compute $q(\hat{x_t}| x_{t+1}, x_{\theta}^{t+1})$, which share the same function format as $q(x_{t-1}| x_{t}, x_{\theta}^{t})$, by substituting with the index $t+1$ and using $\hat{x_t}$ to highlight that it is generated in the sampling stage'
>
>
> **Q7. The exposure bias term $\delta_t$ should be more precisely denoted in the main section. This proposed metric is only introduced in detail in the appendix section.**
>
> **A7**. Thanks for the suggestion, we also realized that the exposure bias $\delta_t$ is important for the readers to understand the metric and we will move it to the main paper.
>
>
>  **Q8. It will be helpful if the Analytic-DPM and the existing exposure bias methods are also considered as the benchmark, even though this does not use the baseline reverse diffusion process by modifying.**
>
> **A8**. We tested the performance of our Epsilon Scaling on Analytic-DPM, PFGM++ even though they are not the initial target baselines of our method (see Table 2 for the results of Analytic-DPM). Regarding the exposure bias solutions, we also directly compare Epsilon Scaling with Input Perturbation [2] (a training regularization to reduce exposure bias) on the baseline ADM. The results shown in Table 4,5 demonstrate the effectiveness and generalization of Epsilon Scaling. We use constant $\lambda_t$ for Epsilon Scaling (ES) below.
>
> Table 4. FID on CIFAR-10 (2nd Heun solver and D=128)
> | NFE | 9 | 13 | 21 | 35 |
> | --- | --- | --- | --- | --- |
> | PFGM++ | 37.82 | 7.55 | 2.34 | 1.92 |
> | PFGM++ with ES | 17.91 | 4.51 | 2.31 | 1.91 |
>
> Table 5. FID on CIFAR-10
> | NFE | 10 | 20 | 50 | 100 |
> | --- | --- | --- | --- | --- |
> | ADM (baseline) | 23.37 | 10.36 | 4.43 | 3.37 |
> | ADM-IP | 20.93 | 6.96 | 2.91 | 2.38 |
> | ADM-ES (ours) | 16.89 | 5.15 | 2.49 | 2.17 |

---

> > ### Comment · Reviewer_nv6a · 2023-11-23
> > **Re: Response**
> >
> > The Question 5 is that, the compared Analytic-DPM method only works fine when the handcrafted heavy clipping of the noise variance at the near-signal part of the sampling phase. If the proposed method works without this clipping, then it will be a better alternative of correcting the posterior variance.
> >
> > Subsequently, we appreciate the authors for detailed response on the questions and concerns that are raised. I remain the current score of marginal above threshold recommending this paper to be accepted, according at the current stage of the revision.

---

### Author Response · Authors · 2023-11-22

Dear Reviewers,

thank you all again for the work you have done and for the feedback you provided. We would like to know if our answers have addressed your concerns or if there is something unclear to you. If you have further questions on any topic, even possibly different from those discussed so far, we would very much like to continue the discussion and provide further details.

---

### Meta-Review · Area_Chair_KrhY · 2023-12-05

**Metareview:**

This paper considers the problem of exposure bias in diffusion models, which is the distribution shift between the forward diffusion and the reverse diffusion outputs. The authors claim that this results in poor synthesis quality. The authors study this phenomenon and propose a training-free remedy called "Epsilon Scaling" to fix this issue.

As pointed by the reviewers, the authors did a good job of motivating the problem. The paper is well written and easy to understand. The experimental results are quite good and they show improved results over several samplers. The tuning of hyper-parameters, however, is still an issue, and in practice, this might take quite some effort. Gains in experimental results are somewhat weak in some cases such as DDIM. However, most reviewers seem to be happy with the rebuttal, and they lean towards acceptance.

Overall, I feel the paper studies an interesting problems which is well motivated. The proposed solution is training-free and seems to work well on most experiments. The contributions would be of good interest to the community, and hence I vote for accepting the paper.

**Justification For Why Not Higher Score:**

Although the paper presents an interesting idea and proposes a simple remedy to fix it, it is not up to the level of a oral paper. Hyper-parameter tuning is one issue that will be of a practical challenge which makes it easy to scale this solution on large-scale. Some more analysis / visualization to understand exposure bias would have made the paper even more stronger.

**Justification For Why Not Lower Score:**

The paper studies an important problem, is well written and has sufficient contributions for an accept. Experimental results sufficiently validate the approach.

---

### Decision · Program_Chairs · 2024-01-16

Accept (poster)